# Exploiting the aggregation propensity of beta-lactamases to design inhibitors that induce enzyme misfolding

Ladan Khodaparast[1,2,4], Laleh Khodaparast[1,2,4], Guiqin Wu[1,2], Emiel Michiels[1,2], Rodrigo Gallardo[1,2], Bert Houben[1,2], Teresa Garcia[1,2], Matthias De Vleeschouwer[1,2], Meine Ramakers[1,2], Hannah Wilkinson[1,2], Ramon Duran-Romaña [1,2], Johan Van Eldere[3], Frederic Rousseau [1,2] ✉ & Joost Schymkowitz [1,2] ✉

There is an arms race between beta-lactam antibiotics development and co-evolving beta-lactamases, which provide resistance by breaking down beta-lactam rings. We have observed that certain beta-lactamases tend to aggregate, which persists throughout their evolution under the selective pressure of antibiotics on their active sites. Interestingly, we find that existing beta-lactamase active site inhibitors can act as molecular chaperones, promoting the proper folding of these resistance factors. Therefore, we have created Pept-Ins, synthetic peptides designed to exploit the structural weaknesses of beta-lactamases by causing them to misfold into intracellular inclusion bodies. This approach restores sensitivity to a wide range of beta-lactam antibiotics in resistant clinical isolates, including those with Extended Spectrum variants that pose significant challenges in medical practice. Our findings suggest that targeted aggregation of resistance factors could offer a strategy for identifying molecules that aid in addressing the global antibiotic resistance crisis.

Bacterial resistance to penicillin was identified even before its introduction for therapeutic use[1] and is thought to have arisen from the selective pressures exerted by β-lactam producing soil organisms[2]. Currently, Extended-Spectrum Beta-Lactamases (ESBLs) have evolved into widespread resistance factors that mediate bacterial tolerance to beta-lactam antibiotics by hydrolysis of the beta-lactam ring, including penicillins, cephalosporins, and to a lesser extent cephamycins and carbapenems[3]. The number of reported beta-lactamases continues to grow rapidly. It currently exceeds over 200 different enzymes[4], and for many of these, there are 10 s or 100 s of closely related variants with an increased activity spectrum[4]. These enzymes are grouped into four amino acid sequence-based Ambler classes, of which A, C, and D use a serine-mediated hydrolysis mechanism, whereas the mechanism of

class B involves a divalent zinc ion[5]. Class A β-lactamases initiate a nucleophilic assault on the β-lactam ring using a water molecule, which is activated by a catalytic serine residue located in the enzyme's active site. This nucleophilic attack results in the breakdown of the β-lactam ring and the creation of an acyl-enzyme intermediate involving the enzyme and the antibiotic[6]. ESBLs represent a particular threat to global healthcare because their resistance spectrum now includes second-, third-, and fourth-generation cephalosporins and monobactams. These strains are also resistant to current β-lactamase inhibitors[2], tazobactam, clavulanate, and sulbactam, which are active-site competitors that form terminal covalent intermediates that inactivate the enzyme[7]. In gram-negative bacteria, notably *Escherichia coli (E. coli)* and *Klebsiella pneumoniae (K. pneumoniae)*, the biggest

[1]Switch Laboratory, VIB Center for Brain and Disease Research, Herestraat 49, 3000 Leuven, Belgium. [2]Switch Laboratory, Department of Cellular and Molecular Medicine, KU Leuven, Herestraat 49, 3000 Leuven, Belgium. [3]Laboratory of Clinical Bacteriology and Mycology, Department of Microbiology & Immunology, KU Leuven, Herestraat 49, 3000 Leuven, Belgium. [4]These authors contributed equally: Ladan Khodaparast, Laleh Khodaparast. ✉e-mail: frederic.rousseau@kuleuven.be; joost.schymkowitz@kuleuven.be

source of ESBLs are the class A beta-lactamases, with over 500 variants of these enzymes recorded in the beta-lactamase database[4]. TEM-1 was the first plasmid-born β-lactamase identified in Gram-negatives, and was found in 1965 in an *E. coli* isolate from a patient in Athens, Greece, by the name of Temoneira[8]. Whereas TEM-1 conferred resistance to penicillin and early cephalosporins the enzyme has demonstrated a striking functional plasticity, adapting the active site to newer beta-lactam antibiotics that were specifically designed to withstand enzymatic hydrolysis. TEM beta-lactamases have spread worldwide and are found in different pathogens, including *Enterobacteriaceae*, *Pseudomonas aeruginosa*, *Haemophilus influenzae*, and *Neisseria gonorrhoeae*[4,9]. SHV-1 (for Sulfhydryl Variable) which shares 68% sequence identity with TEM-1, is chromosomally encoded by the vast majority of the resistant *K. pneumoniae*[10].

Targeting the active site of beta-lactamases as a means of building inhibitors is an attractive idea since even structurally unrelated enzymes that target the same beta-lactam moiety could share some structural similarity in their active site, raising the possibility that a single inhibitor molecule might be effective against a range of target enzymes. Indeed, many beta-lactamase inhibitors, such as tazobactam, are so-called mechanism-based inhibitors that work by competitive active-site binding with a non-hydrolyzable substrate analog. One downside of this approach is the increased selective pressure this exerts on the already fast-evolving beta-lactamase active site. A second and perhaps less appreciated consequence of this approach is that substrate analogs act as pharmacological chaperones[11,12]. A well-known example of such stabilizing substrate analogs in clinical use is deoxygalactonojirimycin (DGJ)[13,14], which is used as a pharmacological chaperone to rescue the folding of alpha-galactosidase in Fabry disease patients and tafamidis, a pharmacological chaperone used for rescuing the folding of transthyretin in familial systemic amyloidosis patients[15]. In particular, the case of DGJ is relevant, since this molecule is an inhibitor that binds in the active site, but when administered at low doses it increases the overall activity of the enzyme by increasing the folding efficiency.

Here, we tested an orthogonal approach for inhibiting beta-lactamase activity, whereby inactivation of β-lactamases is achieved by induced protein misfolding using beta-lactamase targeting Pept-Ins, i.e., synthetic peptides that target aggregation-prone regions (APRs) specific to these enzymes. Importantly these APRs lie outside the active site and are generally conserved in current mutant versions of the same enzyme. Amyloid-like aggregation is an ordered process resulting in an aligned in-register packing of residues in amyloid assemblies. The ordered nature of amyloid-like assemblies also explains why aggregation can be catalyzed by seeding, i.e., by the addition of a sub-stoichiometric amount of pre-formed aggregates, in a manner that is similar to the seeding of crystal growth[16]. Amyloid seeding is sequence-specific, and even a single point mutation is often sufficient to impair seeding between homologs[17]. The vast majority of proteins in any proteome possess at least one APR that can form amyloid-like aggregates. The ubiquity of APRs in proteins is a consequence of globular structure as its tertiary structure requires hydrophobic aggregation-prone sequence segments[18,19]. Intriguingly, most APRs have a sequence that is unique in their proteome[17]. Based on this, the aggregation of any protein could, in principle, be specifically induced by seeding with a synthetic peptide encoding an APR of this target protein. As protein aggregation generally results in protein inactivation, this entails that APRs could therefore be exploited as sequence barcodes directing specific peptide-induced protein functional knock-down. We have recently tested its feasibility in diverse model systems, including prokaryotes[20,21], plants[22,23], and mammalian cells[24].

In this work, we investigated the tendency of TEM and SHV beta-lactamases to aggregate, examining how active-site inhibitors and naturally occurring mutations influenced this phenomenon.

Subsequently, our aim was to create an artificial peptide capable of promoting the aggregation of TEM and SHV beta-lactamases. The objective of this peptide-induced aggregation was to restore susceptibility to beta-lactam antibiotics in clinical isolates that had developed resistance. We evaluated the impact of these peptides on cells and further assessed their effectiveness using a mouse model of bladder infection (Supplementary Fig. 1).

## Results

### TEM and SHV beta-lactamases have a predisposition to misfold and aggregate

We employed the TANGO algorithm[25] to identify all the APRs in the polypeptide sequence of the TEM-1 beta-lactamase. This led to the identification of 7 candidate APRs, ranging in TANGO aggregation score from 11 to 79%, of which one occurs in the signal peptide and 6 occur throughout the globular part of the protein (Fig. 1A and Table 1). To compare this with the distribution of the number of APRs throughout the different structural fold classes, we turned to the SCOP database (release 2.06[26]) and filtered for single chain globular domains and 40% sequence identity using the CD-hit algorithm[27]. This yielded a dataset of 9017 PDB structures of single protein domains divided into 4 roughly equal fold classes: all alpha-helical domains, all beta-sheet domains, domains in which alpha-helix and beta-sheet are intermixed in the sequence, and finally, domains with alpha-helix and beta-sheet separated in sequence. This analysis showed that six APRs is a high number for a protein of this length, although the value is still well within the tail of the distribution (Fig. 1B). Consistent with a high intrinsic aggregation propensity, we observed spontaneous aggregation of the protein in a typical pattern of polar inclusion bodies when we fused TEM-1 to GFP using a linker sequence and expressed it in *E. coli* at 37 °C under an arabinose-responsive promotor (Fig. 1C). When we repeated the same with the homologous SHV-11 enzyme, we made a similar observation (Supplementary Fig. 2B). We recombinantly purified both proteins (without the GFP tag) after overnight expression at 20 °C in *E. coli* BL21 using a pET expression system (Supplementary Fig. 2B, C). Then, we performed a thermal denaturation whilst simultaneously monitoring the intrinsic fluorescence as a measure of the folding status of the molecule (Fig. 1D and Supplementary Fig. 2D) and the static light scattering as a measure of aggregation (Fig. 1E and Supplementary Fig. 2E). This revealed that TEM-1 is only marginally stable, with a melting temperature ($T_m$) of 43.3 °C, whereas SHV-11 has a $T_m$ that is much higher, at 68.8 °C. However, both proteins show a similarly low aggregation onset temperature (Tagg of TEM-1 and SHV-11 is 44.0 °C and 45.6 °C, respectively), consistent with the spontaneous inclusion body formation observed in cells and an overall high aggregation propensity. We confirmed that this did not result from a poor state of the purified material at the onset of the experiment using Size-Exclusion Chromatography (S75, GE Healthcare) coupled to Multiple Angle Light Scattering (SEC-MALS, Wyatt), which showed a single symmetric peak for each protein with a molecular weight that matches that of the monomeric protein within the experimental error of the method (Supplementary Fig. 2B, C). To test the effect of the beta-lactamase inhibitor tazobactam on SHV-11 and TEM-1 stability, we performed the thermal denaturation of TEM-1 and SHV-11 in the presence of an excess of tazobactam and found a decrease in the aggregation of the protein, that was more marked for TEM-1 than SHV-11 (Fig. 1D, E and Supplementary Fig. 2D, E). This confirmed that the beta-lactamase tazobactam acts as a chemical chaperone, the presence of which under appropriate conditions can have the undesired effect of improving the folding of the enzyme, similar to the effect of the alpha-galactosidase inhibitor DGJ described above. To probe whether enzyme stabilization is a general property of active-site beta-lactamase inhibitors, we focused on TEM-1 and performed dose-response experiments for the beta-lactam-based beta-lactamase inhibitors tazobactam (Fig. 1F) and clavulanate (Supplementary Fig. 2F), but also

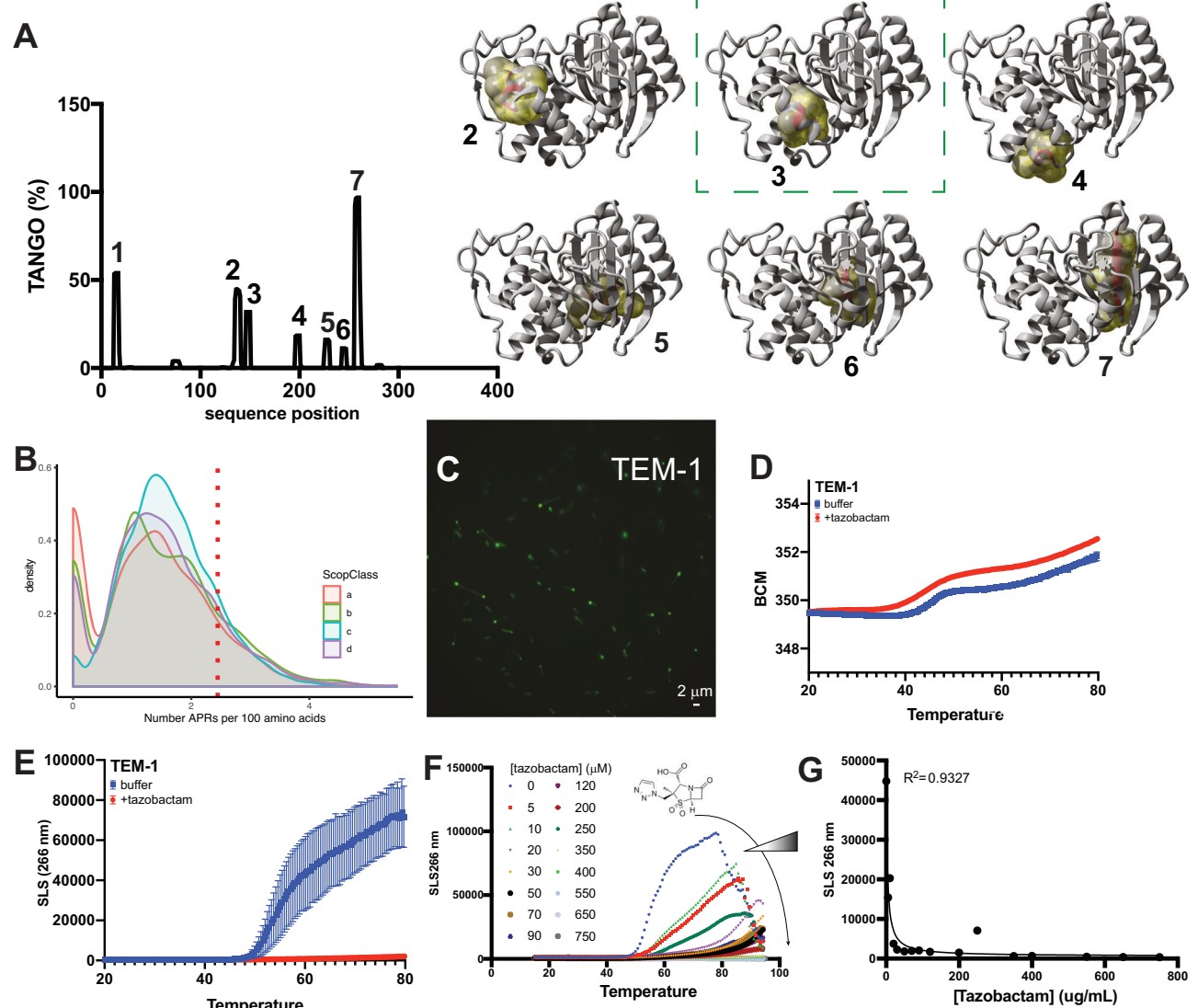

**Fig. 1 | Aggregation propensity of TEM beta-lactamase. A** Aggregation propensity prediction of TEM-1 using TANGO (left) and structural views of the TEM protein with the APR predicted by TANGO highlighted (right). Structure image (not including signal peptide and thus APR1) generated with Yasara of pdb ID 1bt5. **B** Distribution of the number of APRs per 100 residues in the various SCOP categories of protein folds: all-alpha helical (a), all beta-sheet (b), mixed helix and sheet (c), or separate helical and sheet segments (d). The dashed red line indicates the position of the TEM protein. **C** Structured illumination Microscopy (SIM) super-resolution image of *E. coli* BL21 overexpressing a GFP fusion of TEM-1, showing a single representative micrograph of one out of three independent repeats performed. **D** Heat denaturation of TEM-1 monitored by intrinsic fluorescence plotted as the BaryCentric Mean (BCM) of the fluorescence emission spectrum in the presence (red) and absence (blue) of tazobactam. The melting temperature ($T_m$) is derived from these data. The plot shows the mean of 7 replicates, and the error bars represent the standard deviation. **E** Temperature-dependent evolution of the Right-Angle Light Scattering (RALS) intensity measured simultaneously with the data in **D** to monitor protein aggregation. The aggregation onset temperature Tagg is derived from these data. The plot shows the mean of 7 replicates, and the error bars represent the standard deviation. **F** Same data as in **E**, but showing a dose-titration of the beta-lactamase inhibitor tazobactam. **G** Scatterplot of RALS intensity at 60 °C versus the tazobactam concentration (from **F**), also shows a non-linear fit, which has an *R*-squared of 0.93. Source data are provided as a Source Data file.

the non-beta-lactam beta-lactamase inhibitors vaborbactam (Supplementary Fig. 2G) and avibactam (Supplementary Fig. 2H). Although the size and structure of these inhibitors differ widely, all the molecules clearly reduce the aggregation of the enzyme in a dose-responsive manner. Correlation analysis (Fig. 1G) between the amplitude of the static light scattering at 60 °C and the concentration of the inhibitors shows that the relation is statistically significant (*P*-value of the correlation being <0.0001, <0.0004, <0.04, and <0.006 for tazobactam, clavulanate, vaborbactam, and avibactam, respectively). These results show that the pharmacological chaperone effect of active-site beta-lactamase inhibitors is a general property of these molecules, although the magnitude of this effect can vary significantly among the many beta-lactamase−inhibitor pairs.

## APRs have been invariant during TEM evolution

We sought to analyze the effects of mutations frequently found in extended-spectrum TEM variants on the enzyme's stability and aggregation propensity. To do so, we retrieved the TEM variants reported in the Beta-lactamase database (BLDB)[28] and from this set parsed all the mutations identified in clinical samples. We then analyzed their effects on TEM stability using FoldX (pdb-structure 1xpb) and mapped their location to that of the APRs as predicted by TANGO. We also cross-referenced these mutations with earlier published work regarding their effects on TEM enzymatic activity and stability[29,30]. The results of this analysis are shown in Fig. 2A−C. As is clear from Fig. 2A, C, most mutations that extend the TEM spectrum or increase resistance to an inhibitor occur around the substrate-

**Table 1 | APRs identified by TANGO in *E. coli* TEM β-lactamase (UniProt Accession BLAT_ECOLX) and the resulting peptides tested**

| # | Position | APR | TANGO score | Length | Adjusted APR | Peptide sequence |
|---|---|---|---|---|---|---|
| 1 | 12 | FFAAFC | 46.5 | 6 | FFAAFCL | RFFAAFCLRRPRFFAAFCLRR |
| 2 | 134 | LLLTTI | 43.6 | 6 | LLLTTIG | RLLLTTIGRRPRLLLTTIGRR |
| 3 | 145 | LTAFL | 31.9 | 5 | LTAFLHN | RLTAFLHNRRPRLTAFLHNRR |
| 4 | 195 | LLTLA | 18.9 | 5 | LLTLAS | RLLTLASRRPRLLTLASRR |
| 5 | 224 | AGWFIA | 14.5 | 6 | AGWFIA | RAGWFIARRPRAGWFIARR |
| 6 | 242 | IIAAL | 11.1 | 5 | GIIAALG | RGIIAALGRRPRGIIAALGRR |
| 7 | 255 | IVVIYTT | 78.9 | 7 | IVVIYTT | RIVVIYTTRRPRIVVIYTTRR |

binding site and are destabilizing to protein structure. For example, G238S and R164H, two mutations that are very commonly found in clinical samples and are considered to be driver mutations for TEM spectrum extension, are particularly destabilizing to TEM structure. Several compensatory mutations for these destabilizing effects have been described in literature[30], and most of these are indeed predicted by FoldX to enhance protein stability. Strikingly, the lion-share of the mutations observed in clinical samples of extended-spectrum TEM occur outside of the APRs (Fig. 2B). In fact, previous work has revealed that, indeed, the highly evolvable regions in the TEM protein are mostly contained within the flexible loops[31], away from the APRs in the core. It is clear from these observations that the emergence of destabilizing spectrum-expanding mutations in an already marginally stable protein comes with an evolutionary pressure to introduce compensatory stabilizing mutations. Furthermore, likely owing to this marginal stability, the positioning of these mutations seems to be limited to flexible loops and surface positions in the protein, while APRs are largely untouched (Fig. 2C). As such, we reasoned that inducing aggregation of ESBLs through their largely immutable APRs is a viable knock-down approach that cannot be easily escaped through random mutations compatible with enzymatic function.

Of note, the number of amino acid substitutions that separate the vast majority of TEM variants from the original TEM-1 sequence is four or less (Supplementary Fig. 3) out of 263 residues, corresponding to a sequence identity of >98%. Therefore, although these mutations have a major impact on the enzymatic activity because the active site consists of only a few residues, the mutations are not likely sufficient to significantly modify the overall folding and aggregation behavior of the protein. Hence, to develop a sense of the intrinsic aggregation observed for TEM-1, as well as to assess whether the molecular chaperoning effects of the beta-lactamase inhibitors also hold for other TEM variants, we selected representative variants for experimental studies, ensuring to sample the TEM sequence space by including key mutations (highlighted in Fig. 2B). TEM15 (=TEM1[R244S]), TEM30 (=TEM1[E104K;G238S]) and TEM52 (=TEM1[E104K; G238S; M182T]) harbor some of the most common active-site modifying mutations found in ESBLs, including the stabilizing mutation M182T. TEM10 (=TEM1[R165S;E240K]) has been implicated in outbreaks across the US and Europe and TEM155 (=TEM1[Q39K; R164S; E240K]) was included as a more recent extended spectrum, inhibitor-resistant BL. All variants were recombinantly produced as before, and their aggregation was assayed in the presence and absence of tazobactam (Fig. 2D–F and Supplementary Fig. 4). These results lead us to conclude that in the recent evolution of the TEM enzyme, it has retained its intrinsic aggregation propensity and the pharmacological chaperone effect of tazobactam thereon.

### Design and identification of Pept-Ins that inactivate TEM-1

In order to generate a synthetic amyloid peptide capable of inactivating the TEM-1 beta-lactamase enzyme by capitalizing on its aggregation propensity, we applied a previously developed design pattern for synthetic aggregating sequences termed 'Pept-In'. Pept-Ins consist

of a tandem repeat of the APR, each instance flanked by charged amino acids for solubility and separated by a short peptide linker[20,21,24] (Fig. 3A). Previous work has shown that good bacterial uptake can be achieved using one positively charged arginine residue on the N-terminal side of each APR in the tandem and two arginine residues on the C-terminal side[20,21]. This has the additional benefit of reducing the self-aggregation of the peptides and increasing their solubility prior to target engagement. As before, we employed a single Pro residue as a linker between the APRs and focused on an APR length of seven, extending shorter APRs to this length by incorporating flanking amino acids unless they were obvious aggregation breakers (Arg, Lys, Asp, Glu or Pro, see Table 1). We obtained these peptides from solid phase synthesis, followed by HPLC purification to a purity judged by reverse phase chromatography of at least 90% (Genscript). To screen these peptides for their ability to inhibit beta-lactamase activity, we used a clinical isolate of *E. coli* (isolated at University Hospitals Leuven, called UZ_TEM104), which we verified to carry the TEM β-lactamase gene by qPCR and Sanger sequencing and that is highly resistant to penicillin G, showing a minimal inhibitory concentration (MIC) as high as 1600 µg/mL (Table 2) (all clinical isolates are characterized in Suppl Table 2). We determined the MIC of the peptides against this *E. coli* strain and found that they were not toxic up to 100 µg/mL (Table 2). Then, for this strain, we determined the MIC of penicillin G in the presence of the peptides at a fixed concentration of 50 µg/mL (Table 2) and found that the MIC dropped, particularly in the presence of the peptide called TEM3.0 (based on APR 3, RLTAFLHNRRPRLTAFLHNRR), with an 8-fold reduction of the MIC of penicillin G. When we added the widely used β-lactamase inhibitor tazobactam[7] at 50 µg/mL, it reduced the MIC of penicillin G by 4-fold. As expected from the different modes of action of tazobactam and peptide TEM3.0, their combined effect (25 µg/mL of each) reduced the MIC for penicillin G further (to below 50 µg/mL) (Table 2).

### Cross-reactivity with SHV beta-lactamase

Interestingly, peptide TEM3.0 is based on a 5-mer APR (LTAFL, TANGO = 32), which in the Pept-In has been extended to 7 (LTAFLHN) but that is conserved in all 227 TEM sequences in the beta-lactam database[4], and the 7-mer sequence is conserved in 224 out of them (3 carry the H to R mutation in position 6), suggesting that all strains that derive their beta-lactam resistance from TEM1 should be sensitive to treatment with peptide TEM3.0. Moreover, the core sequence of peptide TEM3.0 also occurs in the structurally related SHV β-lactamase (Fig. 3B), although the C-terminal extension residues H and N have been replaced with physicochemically similar R and Q, respectively (Fig. 3C). Hence, to find a peptide with cross-reactivity between the closely related TEM and SHV, we generated TEM3.1, TEM3.2 and TEM3.3, by replacing both or either one of the APR repeats with the version found in SHV (yielding TEM 3.1 RLTAFLRQRRPRLTAFLRQRR, TEM 3.2 RLTAFLHNRRPRLTAFLRQRR and TEM 3.3 RLTAFLRQRRPRLTAFLHNRR, respectively). To quantify the synergy between these four peptides and penicillin G, we performed a so-called checkerboard assay, which allows us to compare the activity of two test compounds

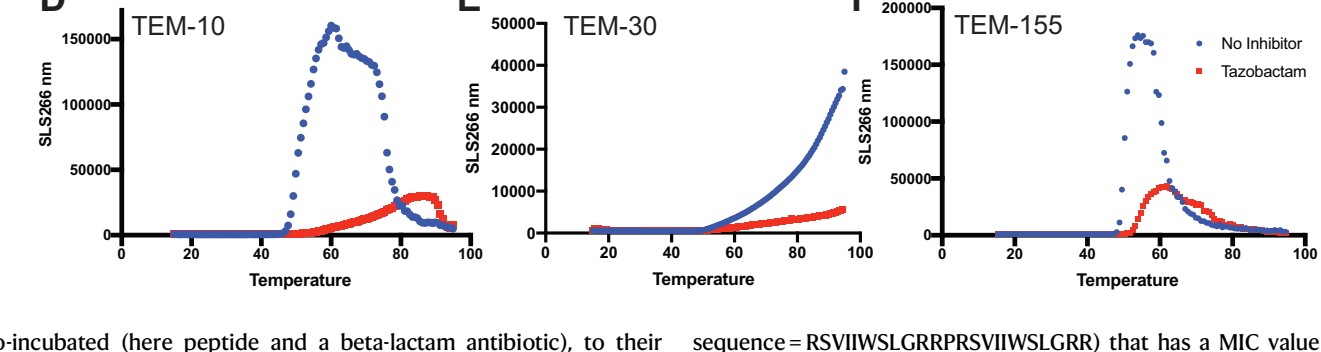

co-incubated (here peptide and a beta-lactam antibiotic), to their individual activities in isolation. The result from the assay is quantified as the Fractional Inhibitory Concentration Index (FICI), and it is generally accepted that values below 0.5 indicate synergistic effects between the compounds. As a control, we generated a peptide using the same design, but based on a previously identified APR from the β-galactosidase enzyme (BGAL)[17] from *E. coli* (APR = SVIIWSL, peptide

sequence = RSVIIWSLGRRPRSVIIWSLGRR) that has a MIC value >100 µg/mL on *E. coli* strain UZ_TEM104. We evaluated the synergy of the peptides with penicillin G in clinical *E. coli* isolates carrying the TEM (Fig. 3D) or SHV enzymes (Fig. 3E). The TEM3.0 peptide, which contains the TEM-version of the APR in both repeats of the tandem design, showed clear synergy only in the strain containing the TEM1 enzyme but not in a strain containing the SHV-11 enzyme. For TEM3.1, which

**Fig. 2 | Effect of mutation on TEM beta-lactamase stability and aggregation.**
**A** TEM mutations and their effects on stability (total energy) as predicted by FoldX. Mutations are categorized according to whether they have been shown to afford resistance to B-lactam antibiotics ("BLACT_RES"), or inhibitors ("INH_RES"), or whether they stabilize protein structure ("stability"). The dashed line indicates a total energy difference of 0.5 kcal/mol, the FoldX cut-off above which mutations are considered to be destabilizing. **B** Barplot mapping mutations in **A** to the TEM sequence and indicating occurrence. *X*-axis shows the position in the primary TEM sequence. The bar heights correspond to the number of times a mutation occurs across different TEM variants (right-hand y-axis). The color coding is identical to that in **A**. The red line indicates the TANGO scores (indicated on the left-hand y-axis), with peaks corresponding to APRs. **C** Mapping of mutations in **A** and **B** to the TEM structure (pdb 1xpb). The protein surface is shown in gray, color coding of mutations is identical to **A** and **B**. **D**–**F** Temperature-dependent evolution of the Right-Angle Light Scattering (RALS) intensity during a temperature ramp, similar to in Fig. 1F for key mutants of TEM1, namely TEM-10 (**D**), TEM-30 (**E**), and TEM-155 (**F**). Experiments were done in triplicate, single repeat is shown. Source data are provided as a Source Data file.

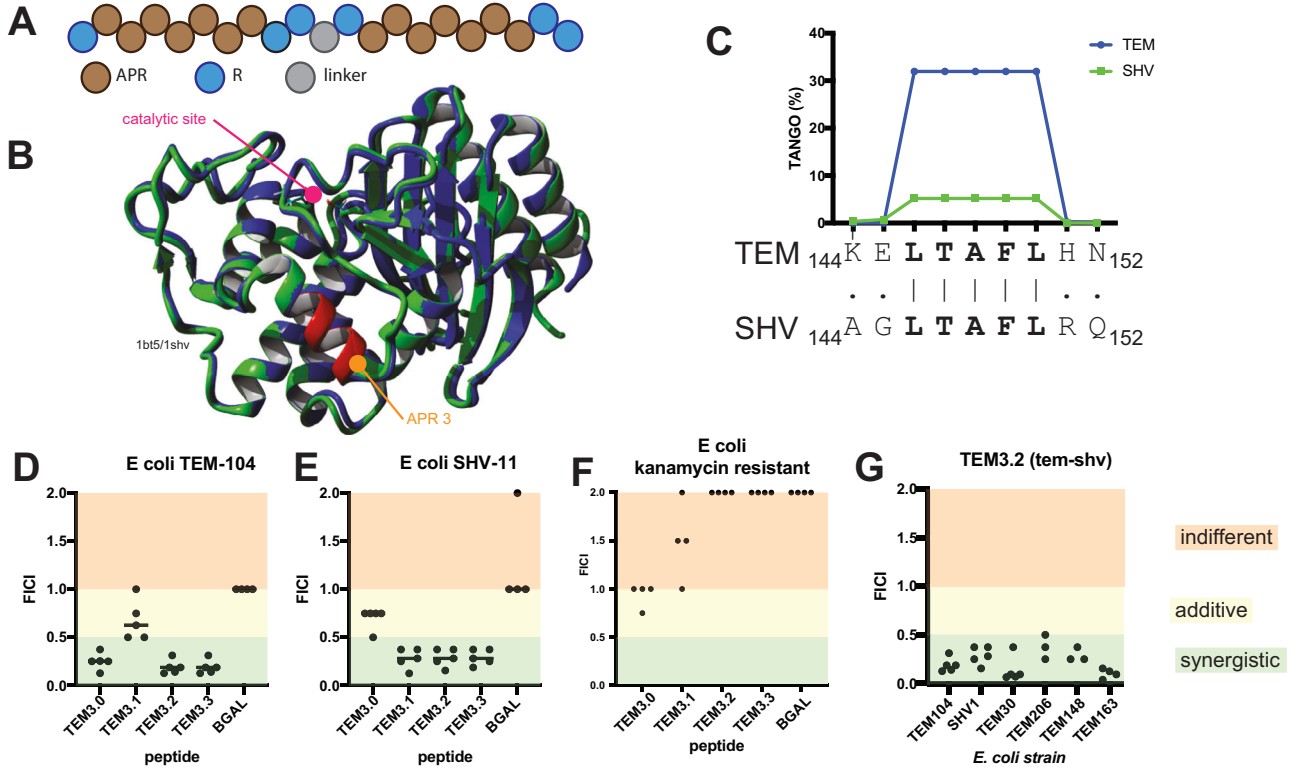

**Fig. 3 | Design and evaluation of peptides that induce aggregation of TEM beta-lactamase.** **A** Schematic representation of the structure of the peptides. APR aggregation-prone region, R arginine. **B** Ribbon representation of the super-position of the crystal structures of TEM (green, pdb id 1bt5) and SHV (blue, pdb id 1shv). The catalytic site of the beta-lactamase activity is indicated and the location of APR3 is shown in red. **C** TANGO aggregation score and alignment of APR3 in TEM and SHV. **D** Fractional Inhibitory Concentration Index determination of the indicated peptides versus the penicillin control on *E. coli* TEM-104. Each of the five dots per condition indicates an independent experiment consisting of 96 data points. FICI values below 0.5 indicate synergy (shaded in green). Values between 0.5 and 1.0 indicate additivity (shaded in yellow) and values greater than 1 indicate indifference (shaded in red) between the combined substances. **E** Same as **D** for *E. coli* SHV-11. **F** Same as **D** for a kanamycin-resistant *E. coli* strain. **G** FICI values for the TEM3.2 peptide on a range of *E. coli* strains. Source data are provided as a Source Data file.

contains the SHV-11 version of the APR in both repeats, showed the opposite: a clear synergy in a strain containing SHV-11 but not in a strain containing TEM1. The TEM3.2 and TEM3.3 peptides, which contain both the TEM and the SHV APRs, showed synergy in both strains containing SHV-11 and TEM-1, suggesting a matching sequence is required for efficient interaction between peptide and protein target. In line with this, the BGAL control peptide showed no synergy with penicillin G, as expected (Fig. 3D, E). As a control, mutant peptides in which proline residues were introduced in the APR regions to break the beta-interactions did not show synergy (Supplementary Fig. 5). As a further control, we also performed a checkerboard assay with an *E. coli* strain that was resistant to the aminoglycoside kanamycin and observed no synergy between this unrelated antibiotic and our peptides (Fig. 3F), in line with the notion that we are inhibiting a beta-lactamase but not the aminoglycoside degrading enzyme (all raw data for the checkerboard assays are in Suppl. Table 3). Then, we further compared the activity of the peptides in 16 additional *E. coli* clinical

isolates containing TEM or SHV enzymes (Table 3), by evaluating the MIC of Ampicillin in the presence of a fixed concentration of 30 µg/mL of TEM3.2. Overall, we found that 87% (14 out of 16) of the clinical isolates containing TEM or SHV were sensitized by the presence of 30 µg/mL TEM3.2 to Ampicillin. The list of strains included some that harbored only TEM or SHV, but others contained additional Ambler Class A (CTX-M or OXA) or Class B (VIM and NDM) beta-lactamases. The CTX-M class of beta-lactamases is a structural homolog to TEM and SHV, and from these data appears to be sensitive to the peptide treatment. The presence of CTX-M and OXA did not appear to modify the effect of the peptide significantly, which may be related to the lower turnover rate of these carbapenemases for Ampicillin used in these experiments[32–35] (Suppl Table 1), but may to some extent also result from indirect effects on their folding and expression due to the proteotoxic stress resulting from the TEM/SHV aggregation. In contrast, the presence of the NDM-type metalloproteases, which have a high affinity for Ampicillin, and are completely distinct in sequence

and structure from TEM and SHV, seemed to completely prevent the effect of the peptide on the beta-lactam sensitivity of the strains in which it occurred (see also further). As it appeared that TEM3.2 had activity in most isolates tested, we decided to focus further analysis on this peptide. Furthermore, we tested the performance of the TEM3.2 peptide in a checkerboard assay on strains carrying ESBL mutants of TEM, i.e., extended-spectrum mutations that cluster around the active site of the enzyme, and not in the APR targeted by the peptide, and as expected from our mode of action, we observed synergy with Ampicillin on these strains (Fig. 3G).

### The Pept-In causes aggregation of the target protein

As expected from the balanced design between aggregation propensity and charge, Dynamic Light Scattering (DLS) showed that the peptide is soluble in vitro, yet aggregates readily when triggered by appropriate interactions, e.g., in the presence of LPS or poly-ionic counterions that are found abundantly inside bacterial cells such as polyphosphate (PolyP) (Fig. 4A), a known modifier of amyloid formation[36] that is upregulated in times of (proteotoxic) stress (Suppl Note 1). Thioflavin T (Tht) binding (Fig. 4B) and Transmission Electron Microscopy (TEM, Fig. 4C, 24 h incubation at RT) revealed amyloid-like

aggregation in the presence of polyphosphate. These fibrils also stain positive for pentameric formyl thiophene acetic acid (pFTAA), another amyloid-specific dye[37–39], that specifically binds to amyloid-like aggregates as well as disease-associated protein inclusion bodies[39] (Fig. 4D). Aggregation-induction by polyP was dose-dependent (Suppl Fig. 6) and could also be induced using agents such as Poly-Ethylene Glycol (PEG) that mimic the molecular crowding that the peptide will encounter in a cellular environment (Suppl Fig. 7). When we evaluated the effect of peptide TEM3.2 on the aggregation of recombinantly purified TEM beta-lactamase protein using a pFTAA aggregation assay, we observed that the peptide was a potent inducer of the aggregation of the TEM beta-lactamase (Fig. 4E–G). This effect was specific as it could not be induced with a Pept-In with a different APR (P2[21]). Importantly, aggregation was associated with a loss of function of the enzymatic function of the beta-lactamase, as assayed using chromogenic β-lactamase substrate nitrocefin (Suppl Fig. 8).

Structured illumination microscopy (SIM) images of *E. coli* UZ_TEM104 treated with a FITC-labeled version of TEM3.2 showed uptake of the peptide into the bacterial cytosol, where it is located in inclusion bodies, the defining organelles of aggregation, that also stain positive for pFTAA (Fig. 5A, B). When we performed the same experiment with an *E. coli* strain lacking TEM (ATCC 25922), we observed only very limited inclusion body formation (Fig. 5C, compared to A), showing that this effect is dependent on the presence of the target protein. Moreover, when we treated bacterial strains stably expressing GFP (Fig. 5D) or proteins fused to FPs (Supplementary Fig. 9) with TEM3.2, we did not observe a similar induced aggregation of the fluorescent proteins, in agreement with the specific induction of aggregation of the TEM beta-lactamase by this Pept-In. In a western blot using a TEM-specific monoclonal antibody of the inclusion body (IB) fraction prepared from *E. coli* UZ_ TEM104, treated with the four TEM peptides, as well as the unrelated BGAL peptide and a buffer control, we observed a clear enrichment of the TEM protein in IBs after treatment with the TEM3.0 (TEM-TEM configuration of APRs), TEM3.2 and TEM3.3 (mixed TEM-SHV APRs), but not for TEM3.1 (SHV-SHV configuration) (Fig. 5E), consistent with the aggregation of the targeted enzyme as intended by the design. The full blot containing control lanes with only peptide or recombinant protein (Supplementary Fig. 10) clearly shows that this band is protein-specific and that there is no cross-binding of the

**Table 2 | MIC values of the TEM peptides as well as penicillin in the presence of peptide or tazobactam for *E. coli* strain UZ_TEM104**

| Peptide | MIC peptide (µg/mL) | Concentration of additive (µg/mL) | MIC penicillin (µg/mL) |
|---|---|---|---|
| None (buffer) | – | 0 | 1600 |
| TEM1.0 | >100 | 50 | 1600 |
| TEM2.0 | >100 | 50 | 800 |
| TEM3.0 | >100 | 50 | 200 |
| TEM4.0 | 100 | 50 | 1600 |
| TEM5.0 | 100 | 50 | 1600 |
| TEM6.0 | >100 | 50 | 1600 |
| TEM7.0 | >100 | 50 | 1600 |
| Tazobactam | – | 50 | 400 |
| Tazobactam + TEM3.0 | – | 25 + 25 | 50 |

**Table 3 | MIC values of penicillin for various clinical isolates of *E. coli* in the absence or presence of 30 µg/mL of peptide TEM3.2**

| Year collected | Country of origin | Infection | Organ | β-lactamase | MIC ampicillin | MIC ampicillin + 30 µg/mL TEM-3.2 |
|---|---|---|---|---|---|---|
| NA | NA | NA | | TEM-1 | >64 | ≤0.06 |
| NA | NA | NA | | TEM-1 | >64 | ≤0.06 |
| NA | NA | NA | | TEM-1 | >64 | ≤0.06 |
| 2016 | South Africa | IAI | Peritoneal Fluid | TEM-ESBL | >64 | ≤0.06 |
| 2016 | South Africa | IAI | Peritoneal Fluid | TEM-ESBL | >64 | ≤0.06 |
| 2016 | Germany | UTI | Urine | TEM-ESBL | >64 | ≤0.06 |
| 2017 | Israel | UTI | Urine | TEM-ESBL | >64 | ≤0.06 |
| 2017 | Portugal | RTI | Bronchi | TEM-ESBL | >64 | 0.5 |
| 2016 | Lithuania | UTI | Urine | TEM-ESBL | >64 | 1 |
| 2017 | Germany | RTI | Bronchi | SHV-12; TEM-ESBL; | >64 | 0.12 |
| 2017 | Turkey | UTI | Urine | SHV-2; CTX-M-55; VIM-1; | >64 | ≤0.06 |
| 2017 | Turkey | IAI | Abscess | TEM-ESBL; CTX-M-24; OXA-48 | >64 | ≤0.06 |
| 2017 | France | IAI | Peritoneal Fluid | TEM-ESBL; CTX-M-TYPE; | >64 | ≤0.06 |
| 2017 | Romania | UTI | Urine | TEM-ESBL; CTX-M-27; | >64 | ≤0.06 |
| 2017 | Qatar | UTI | Urine | TEM-ESBL; CTX-M-15; NDM-5; | >64 | >64 |
| 2017 | Qatar | IAI | Abscess | TEM-ESBL; CTX-M-15; NDM-19; | >64 | >64 |

*UTI* urinary tract infection, *IAI* intra-abdominal infection, *RTI* respiratory tract infection.

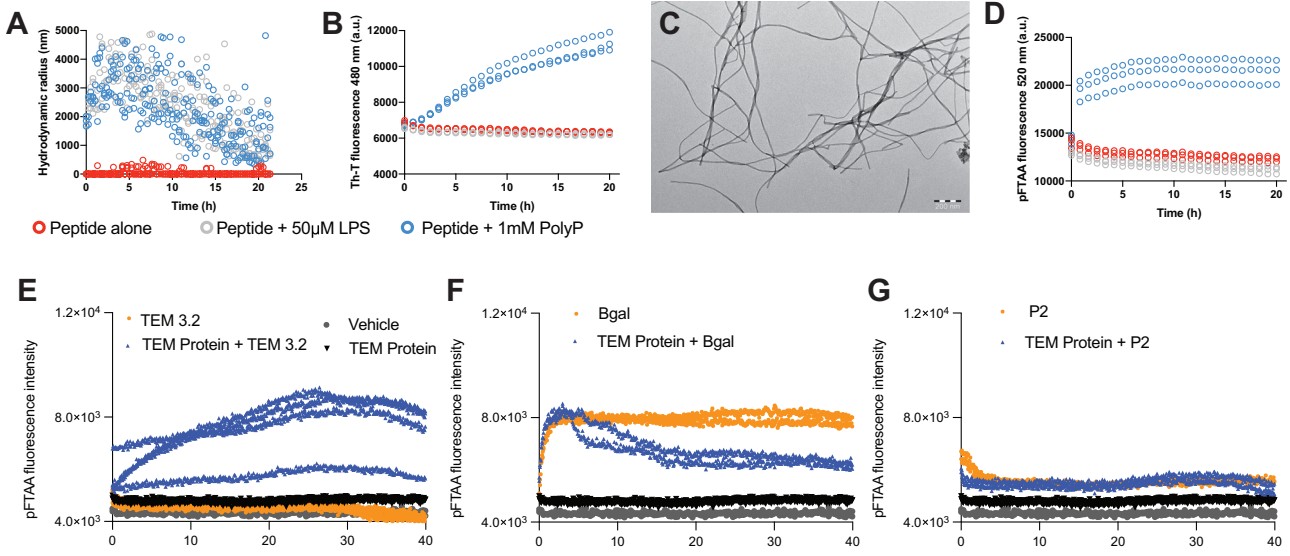

**Fig. 4 | Aggregation kinetics of TEM and SHV proteins and peptides.**
**A** Measurements of the hydrodynamic radius by Dynamic Light Scattering of 50 µM TEM3.2 in buffer alone, or in the presence of LPS or polyphosphate (PolyP). The data show a single representative replicate. **B** Amyloid-like aggregation kinetics of 50 µM TEM3.2 measured by Thioflavin-T (Th-T) fluorescence in the same conditions as **A**. Three replicates are shown for each condition. **C** Transmission Electron Micrograph of 50 µM TEM 3.2 incubated for 24 h in the same buffer as A with polyP, negatively stained with 2% (w/v) uranyl acetate. A single representative image is shown. **D** Amyloid-like aggregation kinetics of 50 µM TEM 3.2 in the PolyP condition in A, measured by pFTAA fluorescence. Three replicates are shown for each condition. **E** Aggregation kinetics using pFTAA fluorescence emission of recombinantly purified TEM-1, in the presence of vehicle, or TEM 3.2 peptin, as well as peptin-only control. **F** Similar as in **E**, but using an off-target Pept-In, based on an APR sequence of beta-galactosidase. **G** Similar to **E**, but now using a previously published off-target Pept-In, based on an APR of HcaB. The plots are the results of a single experiment, all replicates are shown. Source data are provided as a Source Data file.

antibody to the peptide itself. Moreover, we confirmed the presence of the protein in the correct band using mass spectrometry proteomics (Supplementary Fig. 11). We repeated this with a strain expressing SHV-154, blotting with a polyclonal rabbit antibody raised against the recombinant SHV enzyme described above (Fig. 5F). We found that the SHV enzyme accumulates in IBs in response to treatment with TEM3.1, TEM3.2, and TEM3.3, but not TEM3.0, in accordance with the APR configuration in these peptides and reflecting nicely the synergy data mentioned above.

We confirmed these data using FACS analysis of *E. coli* UZ_TEM104 in which we monitored two fluorescence channels: one for Propidium Iodide (PI), a cell death stain that enters cells with an impaired cell wall permeability, and the second for pFTAA, to monitor aggregation in the bacteria. First, we tested the assay using a mixture of live and heat-inactivated (Fig. 6A) or just live bacteria (Fig. 6B), showing that heat-inactivated but not live bacteria are stained with PI, and neither are stained with pFTAA. When we treated bacteria with 400 µg/mL penicillin G only, we observed no major change in either channel (Fig. 6C), consistent with a resistant strain that survives the treatment. When we treated with 50 µg/mL peptide TEM3.2, we observed a shift of the bacterial population only in the pFTAA channel, consistent with non-lethal aggregation occurring in >99% of the cells (Fig. 6D). Finally, when we treated with both 400 µg/mL penicillin G and 50 µg/mL peptide TEM3.2, we observed a shift in both channels, showing that cell death depends on the presence of both penicillin G and TEM3.2 (Fig. 6E).

### Efficacy of TEM3.2 in the treatment of urinary tract infection in mouse

Given its broad activity on clinical isolates, we wondered if the TEM3.2 peptide could be used to re-sensitize bacteria to ampicillin treatment in vivo in a mouse Urinary Tract Infection (UTI) model. Before this, we wanted to assess the potential of the designed peptides to induce off-target aggregation of mammalian proteins, and

we searched the entire human proteome from UniProtKB/Swiss-Prot database (release 2020_04) and found that none of the 20,359 proteins contained an exact match to any of the targeted APRs under study. The same conclusion was reached for the mouse proteome. Then, we established that the peptide was not hemolytic to human erythrocytes (from healthy volunteer donors, Fig. 7A) and not cytotoxic using a Cell Titer Blue assay to human cell lines (in HEK293, HeLa, NCI-H441, SH-SY5Y and HT-1376), as well as primary Human Umbilical Vein Endothelial Cells (HUVEC) and primary mouse cortical neurons (Fig. 7B and Supplementary Fig. 12). We did not observe toxicity of the peptide to mammalian cells in this manner. Moreover, when we treated co-cultures of Hela cell lines and *E. coli* TEM1 with a FITC-labeled derivative of peptide TEM3.2, we observed the fluorescence in the bacterial but not the mammalian cells using fluorescence microscopy, suggesting preferential uptake into the bacteria (Fig. 7C).

We turned to a FITC-labeled derivative of peptide TEM3.2 for the in vivo studies, reasoning that this would allow us to monitor if the peptide could reach the bladder upon parenteral administration using fluorescence imaging, and this was confirmed in a limited study (Supplementary Fig. 13). We then extracted urine from the treated animals 1h after administration and performed SDS-PAGE analysis (Supplementary Fig. 14), which showed the peptide to be largely intact at this stage, allowing us to follow the FITC label. We also performed a tolerability study using a dose escalation method by administering the peptide at 2, 5, 10, 15, and 20 mg/kg via different parenteral routes: intravenous (IV), intraperitoneal (IP), and subcutaneous (SC) in female C57BL/6JAX mice, aged 8–10 weeks (Suppl Table 4). We recorded any clinical signs by looking at their activity, posture, and respiration rate to establish the tolerance of the animals to the substance, and found no obvious signs of toxicity. As a final step, we exposed animals to daily IV injections of TEM3.2 (5 mg/kg) and performed a hematologic analysis, which showed no major discrepancies to vehicle-treated controls (Suppl Table 5). Then we used a catheter to inoculate the urethra of

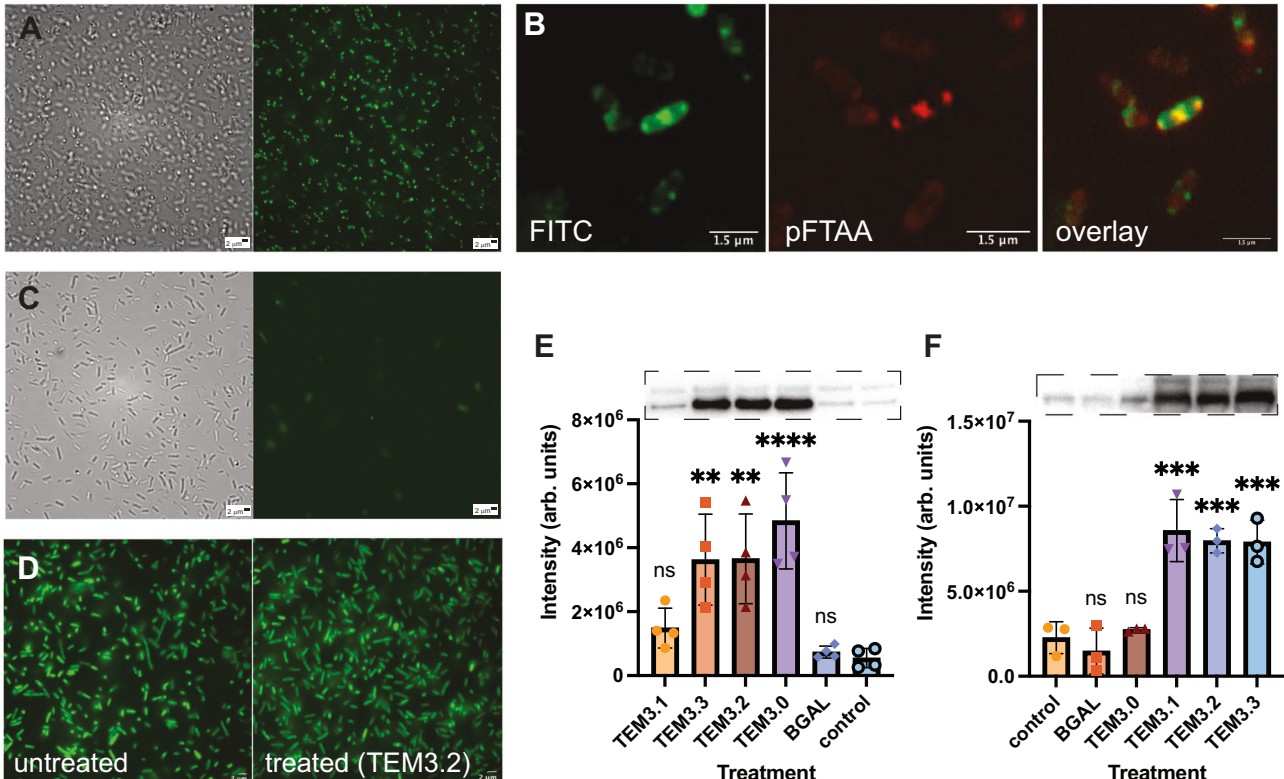

**Fig. 5 | Induction of aggregation in cells. A** Brightfield and structured illumination microscopy (SIM) images of *E. coli* strain UZ_TEM104 treated with TEM3.2 at 12 µM for 120 min and stained with pFTAA. A single representative image is shown. **B** SIM image of *E. coli* strain UZ_TEM104 treated with 12 µM FITC-TEM 3.2 in PBS for 120 min and stained with the red-shifted oligothiophene HS169 to visualize aggregation. A single representative image is shown. **C** SIM and brightfield images as in **A**, but for *E. coli* ATCC strain. **D** SIM images of *E. coli* K12 MG1655 over-expressing GFP from a pBAD vectorn and treated with vehicle or peptide TEM3.2. A single representative image is shown. **E** Western blot and quantification for the TEM beta-lactamase in the inclusion body (IB) fraction of *E. coli* strain UZ_TEM104, treated with 12 µM of the indicated peptide in PBS or control for 120 min. A single

representative blot is shown. The quantification is the result of the densitometric quantification of four independent experiments and shows the mean and the standard deviation. Statistical testing was done using one-way ANOVA with Dunnett's pairwise comparison to control (ns non-significant, *$P ≤ 0.05$, **$P ≤ 0.01$, ***$P ≤ 0.001$, ****$P ≤ 0.0001$). The corrected *P*-values were 0.6091, 0.0029, 0.0027, <0.0001, and 0.9992, from left to right. **F** Western blot and quantification for the SHV beta-lactamase in the IB fraction of *E. coli* SHV-11, treated with 12 µM of the indicated peptide in PBS for 120 min or control. Statistics exactly as in **E**, but from 3 independent experiments. The corrected *P*-values were 0.8753, 0.9796, 0.0001, 0.0003, and 0.0003, from left to right. Source data are provided as a Source Data file.

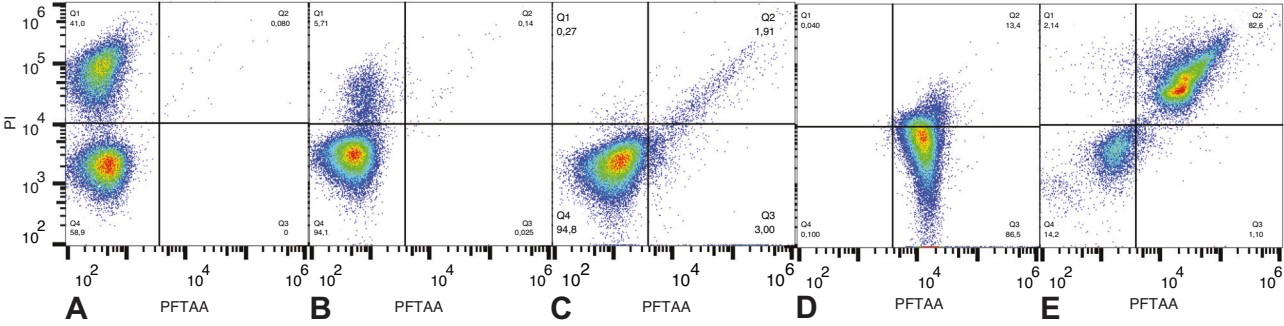

**Fig. 6 | Uptake and inclusion body formation upon peptide treatment.**
Fluorescence-activated cell-sorting (FACS) (**A**) of *E. coli* strain UZ_TEM104 mixed 50–50% with the same strain after heat-inactivation. The cells were stained with pFTAA to monitor aggregation and propidium iodide (PI) to monitor cell per-meabilization associated with cell death. $10^6$ cells were analyzed for the plot. The plot shows a single representative run of three independent repeats. **B** Similar FACS analysis as in A, but with a sample of live bacteria only. **C** Similar FACS analysis as in

A, but for the same *E. coli* strain treated for 4 h with 400 µg/mL penicillin. **D** Similar FACS analysis as in A, but treated with 50 µg/mL TEM 3.2 in PBS for 4 h. **E** Similar FACS analysis as in A, but treated with 400 µg/mL penicillin and 50 µg/mL TEM3.2 in PBS for 4 h. Note: the horizontal and vertical lines correspond to the gatings used to obtain the quantifications shown in the corners of each image. Source data are provided as a Source Data file.

female C57BL/6JAX mice of 8–10 weeks with $1 × 10^8$ cells of a uro-pathogenic *E. coli* strain (UPEC strain blaTEM-1, MIC$_\text{Ampicillin}$ = 1200 µg/mL, which in the presence of 32 µg/mL of either TEM3.2 or tazobactam drops to 25 µg/mL). At 60- and 120-min post-infection, the mice

received 30 mg/kg of Ampicillin orally, plus 10 mg/kg of FITC-TEM3.2 administered IV, IP, or SC, or 10 mg/kg tazobactam, administered IV, as a control (Fig. 7D). Vehicle alone (0.9% NaCl) was also administered (IV). After 24 h the animals were sacrificed and the bacterial load in the

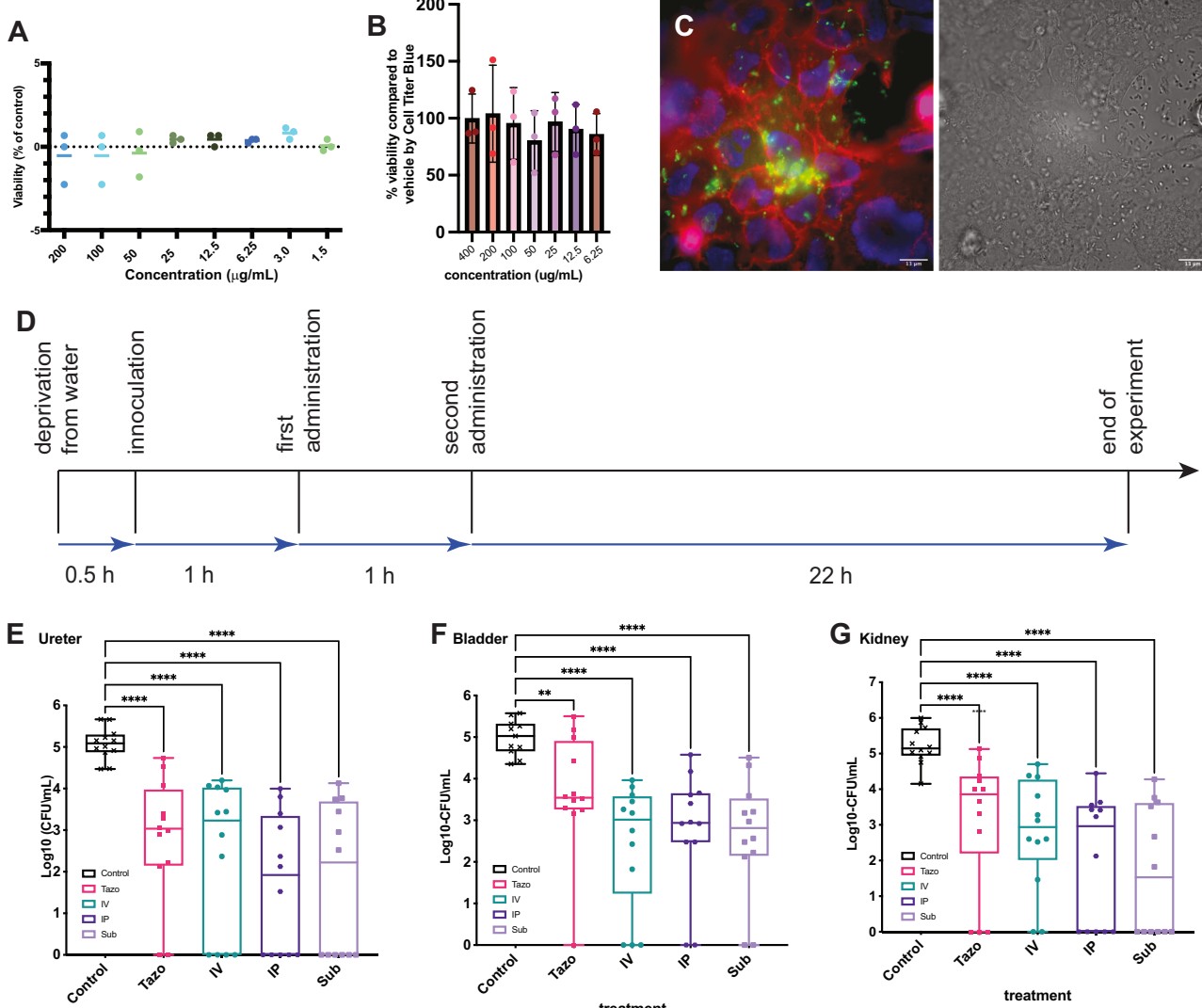

**Fig. 7 | Evaluation of in vivo use of peptides that induce the aggregation of TEM beta-lactamase. A** Lysis of human erythrocytes (hemolysis) was assessed in the presence of the indicated concentrations of TEM3.2 for 2 h at 37 °C, and normalized to the value obtained with 1% of the detergent triton. The plot is the result of 3 replicates and shows the mean and the standard deviation. **B** Cell viability using the CellTiter Blue assay of HeLa cells treated for 24 h with the indicated concentrations TEM 3.2 at 37 °C. The plot is the result of three replicates and shows the mean and the standard deviation. **C** Fluorescence micrography of HeLa cells co-cultured with *E. coli* TEM104 and treated with FITC-TEM3.2 (green channel). DAPI (4′,6-diamidino-2-phenylindole) is blue, while CellMask is red. **D** Experimental workflow of the in vivo model used in this study. **E–G** Bacterial load of indicated organs of female C57BL/6JAX mice with a urinary tract infection with *E. coli* strain UZ_TEM104, treated with 30 mg/kg penicillin (oral) as well 10 mg/kg tazobactam (oral) or 10 mg/kg TEM3.2 via the indicated treatment route. IV intravenous, IP intraperitoneal, SC subcutaneously. The data shown is from a single experiment with 12 animals for each group. The result for each animal is shown as a dot. The plot also shows a box plot, where the box extends from the 25th to 75th percentiles, the line in the middle of the box is plotted at the median and the whiskers go from the minimum to maximum values. The statistically significant differences were determined using ANOVA, followed by Dunnett's multiple comparison test. One control sample for bladder was removed as an outlier (ns non-significant, *$P \le 0.05$, **$P \le 0.01$, ***$P \le 0.001$, ****$P \le 0.0001$). The exact *P*-value in the tazobactam versus control comparison for bladder was 0.0077, for all other comparisons the *P*-value is only known to be <0.0001. Source data are provided as a Source Data file.

bladder, kidney, and ureter was quantified by determining the number of colony-forming units (CFU) per mL of tissue extract (Fig. 7E–G). These graphs showed a reduction of the bacterial load of 2 log folds (up to 2.8), which was most robust in the kidney and ureter, and was slightly outperforming tazobactam (best reduction 1.8 log folds). When we performed FACS analysis similar to Fig. 6A–E on the bacteria from treated, infected animals, we could detect both peptide uptake and protein aggregation in these bacteria, consistent with conservation of our mode of action to the in vivo situation (Supplementary Fig. 15). These data provide proof of concept that the TEM 3.2 is capable of restoring sensitivity to the beta-lactam antibiotic ampicillin in vivo of this strain of beta-lactamase carrying *E. coli*, which could lead to therapeutic applications.

## Targeting the NDM1 beta-lactamase

The sensitivity of clinical strains to the TEM3.2 Pept-In seemed to depend on the presence of other resistance factors since strains carrying the New Delhi Metalloprotease (NDM1) beta-lactamase, which bears no structural or sequence similarity to TEM/SHV, did not respond to TEM3.2 (Table 3). As we wondered about the generality of our peptide design approach, we set out to design a peptide to inactivate a structurally unrelated beta-lactamase. To this end, we turned to the Ambler class B beta-lactamase NDM-1 that confers resistance to many beta-lactam antibiotics, including carbapenems, and strains carrying this enzyme were soon dubbed 'superbugs' that are only partially sensitive to colistin and tigecycline[40]. This enzyme was first isolated from a Swedish patient of Indian descent in 2008[41]. By 2010 it

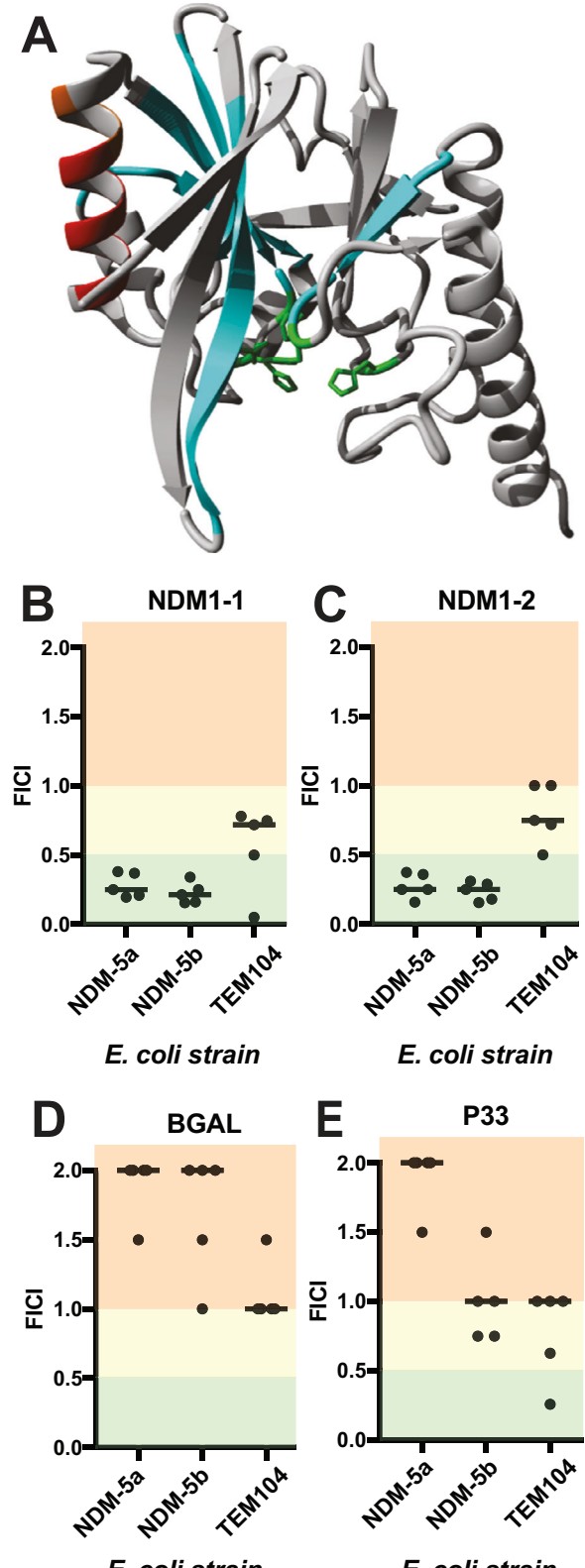

could be detected in clinical isolates throughout the UK and India[42], and by 2015, in over 70 countries worldwide[43]. We set up a screen similar to the one executed for TEM described above, leading to the identification of 2 peptides named NDM1-1 (RTAQILNWRRPRTA-QILNWRR) and NDM1-2 (RLAAALMLRRPRAQILNWIRR) that target the $T_{101}AQILNW_{107}$ APR, detected by the Waltz algorithm[44], located in an exposed alpha-helix in the native structure (Fig. 8A). Both peptides show synergy in strains containing NDM5a and NDM5b (Fig. 8B, C for NDM1-1 and NDM1-2, respectively) but not in a strain containing TEM-1. As controls, we tested a non-toxic peptide on both strains (BGAL, Fig. 8D) as well as a toxic peptide (P33, Fig. 8E), and neither showed synergy on any of the strains used. These results show that targeting beta-lactamases for aggregation could have broader applicability.

## Discussion

The introduction of new antibiotics and the development of resistance against these molecules by the bacteria that are their target constitutes a global arms race that will continue for as long as novel antibiotics are introduced[45]. However, most antibiotics can be classified into a few chemical families, and the need for truly novel chemical classes of antibiotics was dramatically illustrated by the reduced time to resistance of more recently developed molecules, with resistance mechanisms to very similar molecules probably being present in the population prior to market release. The development of non-hydrolyzable substrate analogs of beta-lactamases as drugs to inhibit that particular class of resistance factors could be seen in the same light: by increasing the selective pressure the beta-lactamase active site is under, these drugs may lead to the perverse effects of accelerating beta-lactamase evolution. Therefore, a need exists for alternative strategies, and we here explored the potential of targeted protein aggregation for this purpose. Previously, we had demonstrated the selective targeted aggregation of VEGF receptor 2 in mammalian cells[24] and of the negative regulator of the growth BIN2 in *Arabidopsis* and *maize*[22,23], but in bacterial cells targeted aggregation had thus far only been used to cause widespread aggregation in both gram positive[20] or Gram-negative strains[21], and not selective inactivation of a specific protein. By targeting specific beta-lactamases, we demonstrate the cases of selective inactivation of a bacterial protein via targeted aggregation. The aggregation of TEM, SHV, and NDM is in itself not a lethal event to the bacterial cells but restores sensitivity to beta-lactam antibiotics, indicating that loss-of-function to aggregation is only toxic under conditions where the affected protein is essential for survival, as was previously shown in the mammalian system[24]. Also, the peptides show a striking selectivity between the three analyzed beta-lactamases, as synergy is only observed between each specific pair of peptides and the enzyme it is targeting. The downside of this is that different peptides need to be developed for each beta-lactamase enzyme, but the advantage is that identifying inhibitors for newly emerging enzymes is much faster than identifying novel small molecules. Although the peptides presented in this work have not been optimized in any way to have drug-like properties, they already show efficacy in vivo by restoring sensitivity to beta-lactam antibiotic treatment in a murine bladder infection model. This suggests the development of these molecules into drugs may be possible and that they could become part of the molecular arsenal to combat the global antibiotic resistance crisis.

## Methods

### Bioinformatics analysis

Protein sequences for beta-lactamase TEM, SHV, and NDM bacterial variants were obtained from UniProt[46]. We employed the software algorithm TANGO[25] using default settings to identify APRs across this work, using a score of 5 per residue as the lower threshold and a parameter configuration of temperature at 298 K, pH at 7.5, and ionic strength at 0.05 M. Known TEM variants were retrieved from the Beta-

**Fig. 8 | Peptides that target the NDM1 beta-lactamase. A** Ribbon representation of the crystal structure of the NDM-1 beta-lactamase enzyme (pdb id 3pg4). The active-site residues are indicated in green, the targeted APR is shown in red. **B** FICI plot as described in Fig. 2, for the strains indicated treated with penicillin and peptide NDM1-1. **C** Same as **B** for peptide NDM1-2. **D** Same as **C** for peptide BGAL. **E** Same as **D** for peptide P33. Source data are provided as a Source Data file.

Lactamase DataBase (BLDB)[4]. The mutations found in these variants were cross-referenced with literature to classify them according to their observed effects: offering resistance to an extended spectrum of B-lactams, offering resistance to inhibitors, stabilizing the TEM structure or other[9,47,48]. The effect of each mutant on protein stability was predicted through the FoldX forcefield[49]. To this end, PDB-structure 1xpb[50] was first energy-minimized using the FoldX *RepairPDB* command, and subsequently, the effect on the stability of individual mutations was assessed using the *BuildModel* command, with default settings. As stated above, the TEM sequence was further analyzed using the TANGO aggregation prediction software, using default settings. The mutated residues were visualized in the TEM structure using YASARA[51].

### Peptides design and synthesis

Peptide hits were ordered from Genscript at >90% purity and were also produced in-house using the Intavis Multipep RSi automated synthesizer using solid-phase peptide synthesis. After synthesis, crude peptides were stored as dry ether precipitates at −20 °C. Stock solutions of each peptide were either prepared in 100% DMSO (only for initial screening assays) or following the optimized protocol: peptides were dissolved in 1 M $NH_4OH$, allowed to dissolve for ~5 min, and dried in 1.0 mL glass vials with a $N_2$ stream to form a peptide film. This film was dissolved in a buffer containing 50 mM Tris (pH 8.0) and 20 mM guanidine thiocyanate. Peptides were N-terminally acetylated and C-terminally amidated. Sample peptide QC data are shown in Supplementary Fig. 16.

### Biophysics study

A DynaPro DLS plate reader instrument (Wyatt, Santa Barbara, CA, USA) equipped with an 830 nm laser source was used to determine the hydrodynamic radius ($R_H$) of the peptide particles. Two hundred microliters of each sample (at 100 or 10 μM, unless stated otherwise) were placed into a flat-bottom 96-well microclear plate (Greiner, Frickenhausen, Germany). The autocorrelation of scattered light intensity at a 32° angle was recorded for 5 s and averaged over 20 recordings to obtain a single data point. The Wyatt Dynamics v7.1 software was used to calculate the hydrodynamic radius by assuming linear particles. The amyloid-specific dye Thioflavin-T (Th-T, Sigma-Aldrich, CAS number 2390-54-7) was used to study the aggregation state of peptides. Two hundred microliters of each peptide sample (at 100 μM, unless stated otherwise) was placed into a flat-bottom 96-well microclear plate (Greiner, Frickenhausen, Germany), and the dye was added to a final concentration of 25 μM. A ClarioStar plate reader (BMG Labtech, Germany) was used to measure fluorescence by exciting the samples at 440-10 nm, and fluorescence emission was observed at 480-10 nm (or a complete spectrum ranging from 470 to 600 nm). Aggregation kinetics were obtained by placing 200 μL of the peptide solution with a final concentration of 25 μM thioflavin-T (Th-T) into a flat-bottom 96-well microclear plate. Fluorescence emission was monitored at 480-10 nm after excitation at 440-10 nm. Every 5 min Th-T fluorescence was measured.

### Bacterial collection and growth conditions

Beta-lactamase clinical samples were collected from University Hospitals Leuven and tested for ESBL production using the disk diffusion method[52]. The beta-lactamase reference isolates were purchased from International Health Management Associates (IHMA). Bacterial strains were cultivated in Mueller–Hinton Broth (MHB, Difco) at 37 °C. Whenever required, growth media were supplemented with appropriate antibiotics to the medium or plates (kanamycin 30 μg/mL, L-arabinose 0.5 mg/mL, and IPTG 1 mM/mL). *Escherichia coli* BL21 (ThermoFisher Scientific, Belgium) was used for cloning and plasmid amplification. For the selection of antibiotic resistant colonies, *E. coli* carrying plasmids was grown in LB agar plates supplemented with the relevant antibiotic. Bacterial CFU counting was done on blood agar plates (BD Biosciences, Belgium) or MHA agar plates. Species identification and antibiograms for all clinical isolates were performed using MALDI-Tof and VITEK® 2 automated system (BioMérieux, France). All strains and their resistance profiles are listed in Suppl Table 2.

### In vitro toxicity on the mammalian cells

The Cell Titer Blue assay was performed to evaluate the cell viability according to the instructions of the manufacturer (Promega, USA). The peptide treatments were done in DMEM medium without serum. Briefly, cells were seeded to approximately 20,000 mammalian cells per well in a 96-well flat-bottom plate (BD Biosciences 353075) and incubated at 37 °C with 5% CO2 and 90% humidity. Peptides were diluted in cell medium, and cells were treated for 24 h. 20 μL of the CellTiter Blue reagent was added to each well and the plate was incubated for 1 h at 37 °C. The fluorescence was measured at 590 nm by exciting at 560 nm with a ClarioStar plate reader (BMG Labtech, Germany).

Hemolytic activity was evaluated by measuring the amount of released hemoglobin. Fresh blood was pooled from healthy volunteers (collected from Rode Kruis Vlaanderen, Mechelen, Belgium). EDTA was used as the anticoagulant. Briefly, erythrocytes were collected by centrifugation $3000 \times g$ for 10 min. The cells were washed with phosphate-buffered saline (PBS) several times and diluted to a concentration of 8% in PBS. 100 microliters of 8% red blood cells solution were mixed with 100 μL of serial dilutions of peptides in PBS buffer in 96-well plates (BD Biosciences, Belgium). The reaction mixtures were incubated for at least 1 h at 37 °C after which plates were centrifuged for 10 min at $3000 \times g$. The release of hemoglobin was determined by measuring the absorbance of the supernatant at 495 nm. Erythrocytes lysed in 1% Triton were used as control for 100% hemolysis.

### Antibody and antibiotic product codes

The antibodies and antibiotic product codes used are as follows: monoclonal anti-TEM (Abcam, UK ab12251-8A5A10) 0.5 μg/mL, polyclonal rabbit anti-SHV (custom-made by Eurogentec, Belgium) 1 μg/mL, chicken polyclonal anti-beta Galactosidase (Abcam, ab145634 antibody (ab9361) 2 μg/mL. Goat Anti-Mouse IgG HRP secondary antibodies (ab97040); Rabbit Anti-Mouse IgG HRP (ab6728); Goat Anti-Chicken HRP (ab97135).

The antibiotics used for this study: Penicillin G sodium (Benzylpenicillin sodium, Abcam, catalog # ab145634) 1 μg/mL, Ampicillin (Duchefa Biochemie, Netherlands, A0104.0025), tazobactam sodium salt (Sigma-Aldrich, catalog # T2820-10MG), erythromycin, CAS number 114-07-8 (Sigma-Aldrich, catalog # E5389), chloramphenicol, CAS number 56-75-7 (Duchefa Biochemie), and kanamycin CAS number 56-75-7 (Duchefa Biochemie).

### MIC determination

Determination of MIC values was performed using the broth microdilution method according to the EUCAST guideline, which was performed in 96-well polystyrene flat-bottom microtiter plates (BD Biosciences). Briefly, a single colony was inoculated into 5 mL Difco™ Mueller−Hinton Broth (BD Biosciences Ref 275730) and grown to the end-exponential growth phase in a shaking incubator at 37 °C. Cultures were subsequently diluted to a MacFarland (0.5 optical density) to reach $10^6$ CFU/mL in fresh MHB medium. 50 μl of different concentrations of peptides ranging from 128 to 2 μg/mL were serially diluted to the sterile 96-well plate in MHB. 50 μL of the diluted bacteria in MHB were next pipetted into 96-well plates to reach the final volume of 100 μL. The bacteria grown with the maximum concentration of carrier and medium were considered positive and negative controls, respectively. The plates were statically incubated overnight at 37 °C to allow bacterial growth. OD was measured at 590 nm using a multipurpose ultraviolet−visible plate reader, and the absorbance of the

bacterial growth was measured using an absorbance reader. Bacterial growth was also visually inspected which agreed well with the OD reading.

### β-lactamase activity assays

The beta-lactamase assay procedure is based on the hydrolysis of the substrate Nitrocefin, a chromogenic cephalosporin, which produces a colored product (detectable at OD = 490 nm) that is directly proportional to the quantity of beta-lactamase activity. This experiment was carried out in 96-well black polystyrene flat-bottom microtiter plates (BD Biosciences). In brief, 50 μl of different concentrations of peptides or tazobactam in PBS were added to each well, followed by 50 μl of beta-lactamase protein at a final concentration of 12 ng. The plate was incubated at 37 °C for 1 h. The control heat was heated for 1 h at 95 degrees Celsius. Each well received 5 μl of Nitrocefin at a stock concentration of 0.5 mg/mL Nitrocefin. The hydrolyzed Nitrocefin was identified by absorption at 490 nm, which is proportional to the amount of beta-lactamase activity.

### Checkerboard assay

For the analysis of synergy between the peptides and other antibiotics, a checkerboard assay was performed. Based on the MICs of the selected peptides, a checkerboard assay was designed to define their FICIs (Fractional Inhibitory Concentration Index) in combinations against different clinical isolates[53,54]. Briefly, a total volume of 100 μL of Mueller–Hinton broth was distributed into each well of the 96-well plates. The first compound (peptide) of the combination was serially diluted vertically (128, 64, 32, 16, 8, 4, 2, 0 μg/mL) while the other drug (Beta-lactam or Kanamycin) was diluted horizontally in a 96-well plate (from 3200 to 3 μg/ mL). The total volume of each microtiter well was inoculated with 100 μL of MHB containing $1 \times 10^6$ CFU/mL bacteria. The plates were incubated at 37 °C for 24 h under aerobic conditions without shaking. Calculation of the FICI is used to analyze the results of the checkerboard assay by estimating the degree of synergistic effect. FICI is calculated as the sum of the individual fractional inhibitory concentrations (FICs) for each drug (where MIC A and MIC B denote the MIC of each drug alone, and MIC $A_{A+B}$ and MIC $B_{A+B}$ denote the concentrations of A and B in the drug combination). FICI = (MIC $A_{A+B}$/MIC A) + (MIC $B_{A+B}$/MIC B). With FICI ≤ 0.5, the combination of antibiotics is considered as a synergistic effect, 0.5 < FICI ≤ 1 indicates additivity, FICI > 1 indicates indifference.

### Flow cytometry analysis

Bacterial cells in cleaned suspensions were stained with both propidium iodide (PI) and FITC-labeled peptides to evaluate the killing rate and peptide uptake in a two-dimensional analysis. Briefly, end-exponential growth phase *E. coli* cells ($10^6$ CFU/mL) were washed with PBS and treated with peptides at sub-MIC (0.25 x MIC) and sub-MIC of Penicillin for several hours at 37 °C. Treated bacteria were washed with PBS buffer two times. One microliter of PI (Invitrogen) was added to the bacteria and incubated for 5 min. The bacteria were counted by FACS to reach 40000 events. To correlate the activity of the peptides with cell death, the fluorescence intensity was measured in two channels using the Gallios™ Flow Cytometer (Beckman Coulter, USA), PI: excitation 536 nm and emission 617 nm, FITC: excitation 490 nm and emission 525 nm. Heated bacteria at 90 °C for 10 min were used as PI-positive control.

### Staining with luminescent conjugated oligothiophenes

The bacterial cultures were washed with PBS, and the number of bacteria was adjusted to $10^8$–$10^9$ cells/mL. Bacteria were then treated with peptides (at sub-MIC or MIC concentration based on the aim of the study) or buffer for 2 h at 37 °C. Then, cells were treated with LCO

dyes (pFTAA; Amytacker™680 or Amytacker™545: final concentration of 0.5 μM; Ebba Biotech, Sweden) for at least 90 min. The absorption, emission, and excitation spectra for each dye were measured based on the standard Ebbabiotec advice (ebbabiotech.com).

### Inclusion body (IB) purification

Overnight cultures of bacteria were centrifuged for 30 min at $4000 \times g$ and cells were washed with physiological water (NaCl 0.9%). Bacterial cells were treated by peptide at the appropriate concentration for at least 2 h at 37 C. The bacterial pellets were washed with 10 mL buffer A (50 mM HEPES, pH 7.5, 300 mM NaCl, 5 mM β-mercaptoethanol, 1.0 mM EDTA) and centrifuged at 4 °C for 30 min at $4000 \times g$. The supernatant was discarded and 20 mL of buffer B (buffer A plus 1 tablet of the protease and phosphatase Inhibitor Cocktail (ab201119, Abcam, UK) was added to the bacterial pellet. In order to break the cells, a High-Pressure Homogenizer (Glen Creston Ltd) with the pressure set to 20,000–25,000 psi was used on ice, and in addition, the suspensions were sonicated (Branson Digital sonifier 50/60 Hz) on ice with alternating 2 min cycle (15 pulses at 50% power with 30 s pauses on ice, until completing 2 min total sonication time). The lysed cells were centrifuged at 4 °C for 30 min at $11,000 \times g$. The precipitated fraction was afterward resuspended with 10 mL buffer D (buffer A plus 0.8% (V/V) Triton X-100, 0.1% sodium deoxycholate), and the suspension was sonicated to ensure the pellet was completely dissolved. This step was repeated three times. Centrifugation was performed at 4 °C for 30 min at $11,000 \times g$. Finally, to solubilize IBs, the pellet was suspended in 500 μl of buffer F (50 mM HEPES, pH 7.5, 8.0 M urea).

### SHV and TEM protein purification

Plasmids were obtained from Genscript (USA) vector construction services. TEM (870 bp) and SHV (894 bp) were each sub-cloned into a PUC57 vector cloning site NdeI/ XhoI, with an N-terminal HIS-tag followed by the TEV cleavage site. The proteins were expressed in *E. coli* BL21 (DE3) by inducing with 1 mM IPTG overnight at 20 °C. Cells were harvested by centrifugation (15 min at 5000 rpm (2800 x g) at 4 °C), resuspended in buffer (500 mM Sucrose, 200 mM Tris pH 8.5 plus protease inhibitors (mini ETDA free (Sigma-Aldrich), one tablet per 25 mL of buffer) and lysed using a high-pressure homogenizer (EmulsiFlex C5, Avestin, Canada). The cell debris was removed by centrifugation (30 min, 18 x g), and the soluble lysate was loaded on a size-exclusion chromatography (SEC) column 26/600 75 pg column (column vol 320 mL, GE Healthcare, USA). The protein was equilibrated with buffer 50 mM Tris pH 8.5, 300 mM NaCl.

### GFP fusion protein construction

TEM- and SHV-GFP fusion proteins were sub-cloned into the Invitrogen pBAD myc/his A vector. To this end, a vector expressing GFP with a linker (sequence KPAGAAKGG) at its C-term, designed in a previous study[55], was modified. A multiple cloning site containing EcoRI and SpeI restriction sites was introduced C-terminally of the linker sequence through site-directed mutagenesis (using the New England Biolabs Q5® Site-Directed Mutagenesis Kit). Next, SHV and TEM sequences with EcoRI and SpeI restriction sites at their N- and C-terminus, respectively, were produced through PCR amplification from the expression constructs used for purification (discussed above). Finally, both the vector and PCR inserts were digested with SpeI-HF® and EcoRI-HF® (New England Biolabs) and ligated according to the manufacturer's instructions.

### Expression of TEM or SHV fusion GFP in *E. coli*

For protein expression and solubility analysis, bacterial strains were grown overnight in Lysogeny Broth (LB Difco™) supplemented with Ampicillin for GFP expression and both Ampicillin and chloramphenicol for co-expression of the GFP constructs with pKJE7. The

overnight cultures were diluted 1:100 in fresh LB supplemented with the appropriate antibiotics and grown to an OD of about 0.6, after which expression was induced with 0.2 % arabinose. Expression was allowed to proceed for 3 h after which cells were lysed in B-PER™ reagent (ThermoFisher, USA) supplemented with 0.1 mg/mL lysozyme (Sigma-Aldrich), Complete™ Protease Inhibitor Cocktail (Sigma-Aldrich) and Pierce™ universal nuclease for cell lysis (ThermoFisher). Cells were lysed on ice for 30 min, after which soluble and insoluble fractions were separated through centrifugation at 17,100 x $g$ for 30 min at 4 °C. The supernatant was removed, and the insoluble fraction dissolved in an equal volume of 8 M urea. GFP in soluble and insoluble fractions was then quantified through SDS-PAGE followed by Western blotting. Blots were developed using chemiluminescence after incubation with an anti-GFP antibody (Antibody 2555 S, Cell Signaling Technologies) or anti-DnaK antibody (D8076, USBio USA) and an HRP-conjugated secondary antibody. Blots were quantified using Bio-Rad's Image Lab™ Software. Soluble GFP fractions were determined by calculating the ratio of soluble over total (soluble + insoluble) protein.

### Experimental animals

Female C57BL/6Jax mice of 6 to 8 weeks with uniform weight (between 20 and 23 g) were used in this study (Harlan, The Netherlands). Mice were housed in plastic cages, four mice per cage on softwood granules as bedding. The room was kept between 21 °C and 25 °C with 12/12 h light–dark cycles. The animals had free access to water and pelleted rodent food. To avoid stress-induced confounding factors, the mice were transferred to the lab one week before experimental manipulation.

### Efficacy of TEM3.2 in the treatment of urinary tract infection in mouse

To test the efficacy of the peptides, a urinary tract infection model was performed as described previously[56]. Briefly, female C57BL/6Jax mice female mice were deprived of water for at least 1 h. Then, they were anesthetized by IP administration of the mixture of ketamine (Nimatek)/xylazine (XYL-M 2% BE-V170581). The bladder of the mouse was massaged with fingers and pushed down gently to expel the remaining urine. Mice were slowly inoculated urethrally with 50 μL of a bacterial suspension slowly over 5 s to avoid vesicoureteral reflux ($10^8$ CFU/mouse) using a sterile catheter (pediatric intravenous-access cannula (GS391350)). The catheter was removed directly after inoculation. After surgery, the animals were visually monitored for full recovery. After 1 h post inoculation, all mice received Ampicillin (30 mg/kg PO-orally) and at the same time 3 groups of animals received the peptides via different administration routes (10 mg/kg IV−intravenous; IP−intraperitoneal or SC−sub-cutaneous) and the positive control groups received tazobactam (10 mg/kg, PO). The negative control groups received vehicle or saline (IV administration). 2 h post inoculation, all mice received a second injection with the same concentration of each treatment as explained above. Twenty-four hours post-infection, mice were sacrificed and organs (kidney, bladder, ureter) were washed with PBS and were homogenized (Thermo Savant FastPrep FP120 Homogenizer/24 s). The homogenized tissues were serially diluted and cultured on blood agar plates. The plates were incubated overnight at 37 °C, and the number of bacteria was measured by CFU value.

### Fluorescence microscopy of co-cultures of bacteria and mammalian cells

Human HeLa cells were grown to create a confluent monolayer on a small-cell-view cellular plate with a glass bottom (Greiner Bio-One GmbH/35 mm Ref: 627860) for imaging purposes. Next, cells were inoculated for 24 h with 200 μL of a mixture of overnight culture of TEM1 *E. coli* strain and FITC-Peptides (3xMIC). Cells were stained for 30 min with CellMask Deep Red plasma membrane dye (ThermoFisher

catalog # C10046) and 1 L of NucBlue reagent (Invitrogen), after which the medium was removed, and 2 mL paraformaldehyde 4% was added to the dish for fixation. The dish was kept at room temperature for 6 h. Prior to imaging, the co-cultured cells were rinsed at least three times with 1 mL PBS.

### Structured illumination microscopy (SIM)

Bacteria were fixed by adding 2.5% paraformaldehyde and 0.04 % glutaraldehyde (final concentrations) to the culture media, followed by incubation at room temperature for 15 min and 30 min on ice. Bacteria were then washed in PBS and resuspended in GTE buffer (50 mM glucose, 25 mM Tris, and 10 mM EDTA, pH 8.0). Directly preceding microscopic analysis, cells were transferred to a glass slide and covered with a coverslip. Imaging was performed using a Zeiss Elyra S.1 system in the VIB BioImaging Core at KU Leuven.

### Statistics

Statistical analysis was performed with Prism or R, using unpaired student's $t$-tests, one-sample $t$-tests, and ANOVA to determine the statistical significance of differences between samples unless otherwise indicated. Significance levels: * for $P < 0.05$; ** for $P < 0.01$; *** for $P < 0.001$; **** for $P < 0.0001$. Non-significant differences are not separately labeled, unless stated otherwise.

### Animal experiments

All mouse experiments were conducted according to the national (Belgian Law 14/08/1986 and 22/12/2003, Belgian Royal Decree 06/04/2010) and European (EU Directives 2010/63/EU, 86/609/EEG) animal regulations. All protocols were approved by the KU Leuven Institutional ethics committee on animal experimentation. All relevant animal characteristics and housing conditions are specified in the materials and methods.

### Patient samples

All blood samples were obtained from healthy volunteers from the biobank of the Red Cross Flanders in accordance with all relevant national legislation, including informed consent. Blood samples were completely anonymized prior to transfer to our facilities. Ethical approval was obtained from the medical ethical committee of the University Hospitals Leuven (study number S60497).

### Cell lines

Cell lines were verified by Eurofins Genomics Europe (accredited acc. to DIN EN ISO/IEC 17025:200). Genetic characteristics were determined by PCR-single-locus-technology. 16 independent PCR-systems D8S1179, D21S11, D7S820, CSF1PO, D3S1358, TH01, D13S317, D16S539, D2S1338, AMEL, D5S818, FGA, D19S433, vWA, TPOX, and D18S51 were investigated. Results were compared with the online database of the DSMZ (http://www.dsmz.de/de/service/services-human-and-animal-cell) and the Cellosaurus database (https://web.expasy.org/cellosaurus. Only the PCR systems with ANSI/ATCC standard ASN-0002 were aligned in the final comparison.

### Reporting summary

Further information on research design is available in the Nature Portfolio Reporting Summary linked to this article.

## Data availability

The datasets generated during and/or analyzed during the current study are available from the corresponding authors. Protein sequences for beta-lactamase TEM [https://www.uniprot.org/uniprotkb/P62593/], SHV [https://www.uniprot.org/uniprotkb/P0AD64/entry], and NDM [https://www.uniprot.org/uniprotkb/C7C422/] beta-lactamases were obtained from UniProt. Known TEM variants were retrieved from the Beta-Lactamase DataBase (http://bldb.eu/alignment.php?align=A:TEM).

We used the following PDB files: 1BT5 (TEM-1), 1XPB (TEM-1), 1SHV (SHV-1), 3PG4 (NDM-1). Source data are provided in this paper.

## Code availability

The code of all analysis scripts and in silico datasets used are available from the corresponding authors upon reasonable request.

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

## Acknowledgements

The Switch Laboratory was supported by grants from the European Research Council under the European Union's Horizon 2020 Framework Program ERC Grant agreement 647458 (MANGO) to J.S., the Flanders Institute for Biotechnology (VIB), KU Leuven ("Industrieel Onderzoeksfonds", C24/17/075 to F.R.), the Funds for Scientific Research Flanders (FWO, project grants G0C3522N and G045920N, and postdoctoral fellowships 1231021N to Ladan K. and 12S3722N to B.H.) and the Flemish Agency for Work and Innovation (VLAIO, Innovation mandate HBC.2020.2854 to E.M.). We thank the following core facilities for training, technical support, and access to their instrument parks: the VIB BioImaging Core at KU Leuven (SIM microscopy), the KU Leuven Flow and Mass Cytometry Facility, and the Electron Microscopy core of VIB-KU Leuven. We thank Bernard Scorneaux, formerly of Aelin Therapeutics, for helpful suggestions and Dr. Stefanie Desmet (University Hospitals, Leuven) for kindly providing clinical isolates.

## Author contributions

Ladan K., Laleh K., J.V.E., F.R., and J.S. conceived and planned the experiments. Ladan K., Laleh K., G.W., E.M., R.G., T.G., & M.R. carried out the experiments. B.H., R.D.-R., and J.S. performed computational analyses. T.G., M.D.V., and H.W. contributed to sample preparation. Ladan K, Laleh K. R.G., J.V.E., F.R., and J.S. contributed to the interpretation of the results. J.S. and F.R. took the lead in writing the manuscript. All authors provided critical feedback and helped shape the research, analysis, and manuscript.

## Competing interests

Ladan K., Laleh K., J.S., and F.R. are named as inventors in a patent (WO 2022/184821, 'Beta-lactamase inhibitors') filed by their host institute VIB describing the peptides mentioned in this manuscript (Status: international PCT phase, pending non-licensed). The remaining authors declare no competing interests.
