## [Peer Review File · Nature Communications]

REVIEWER COMMENTS

Reviewer #1 (Remarks to the Author):

The authors aim to show the efficacy of the TEM3.2 in the treatment of UTI, to eliminate bacterial growth from the urinary tract. As suggested, I have focused on the mouse model and the rest I have only glanced at. Therefore, misunderstandings might have happened.

The mouse model is commonly used by us and others. One can of course argue that more sophisticated analysis could have been done, but I think the only thing that was relevant to them were the bacterial count.

Still, I have some comments on the experimental and associated parts. Please see below.

Results:

Line 373: The authors comment different cell-lines used to show that the peptide is not cytotoxic. A wide range of cells have been tested. However, I have problems understanding why these specific cells were chosen and the immediate relevance for their study. The cells were derived from i.e. cervical cancer, distal lung epithelial cells and neuroblastoma. I lack bladder cells.

Line 378: The authors mention that they treat cell lines, but it is not stated which cells and unfortunately, I am therefore not sure if it really is the HeLa cells as indicated in the Figure legends. I would suggest using a bladder cell line instead, since this section is about UTI.

Line 386: The dose response curve mentioned in the text would fit well in the Supplement and would improve the understanding for the reader.

Line 396: Were mice inoculated with both E coli blaTEM-1 and at the same time either tazobactam or TEM3.2?

Line 396: At 60 and 120 min respectively mice were treated with ampicillin + FITC TEM32 (? Is it a typo and should instead be TEM3.2?) given either IV, IP or SC and as control tazobactam. To pretreat at the same time as start of infection is often done but is unfortunately far from real life. I would therefore recommend treating only after 60 and 120 min, especially as the authors claim it could be a therapeutic alternative (Line 409).

Figure 4: The labeling has drifted and should be adjusted. The figure legend is wrong and indicates that the bacterial growth is shown in figures C,D,E, but should be D,E,F.

Figure 4. Only 4 mice were included in each group. Although there are significances, I suggest adding a few more mice in each group to secure the results, since there is sometimes a big variation in results.

Figure 4. It might be obvious to everyone, except for me, but what is LOD as written in the figure 4 D,E,F.

Reviewer #2 (Remarks to the Author):

Review:

The manuscript entitled “Exploiting the aggregation propensity of beta-lactamases to design inhibitors that induce enzyme misfolding” (NCOMMS-23-00655) by Schymkowitz, Rousseau and others focuses on the aggregation of β -lactamases, a group of enzymes that degrade β -lactam antibiotics and considered the major contributors for antibiotic resistance. This work elucidates the effect of β -lactamases aggregation on their activity and resistance to antibiotics. The paper is interesting and important, both from basic perspective and its applications in fighting biofilm resistance to antibiotics. Yet, the manuscript should be edited to better clarify the take-home messages and to explain it to a wider audience. In addition, several more controls are needed to elaborate on the chemical mechanism behind the described phenomenon and inhibitors. Most importantly, few of the results appearing in the figures should be removed to the supplemental section, in order to keep the manuscript focused and clear. Specific comments:

Major revisions:

1. Figure 1 – the figure is too dense and contains a lot of data that is not crucial for understanding the main message and can be removed to the supplementary information section (for example, Figure 1D is nice and impressive but not necessary in the figure and should be only mentioned in the text). Also, too many concentrations are showed at Figure 1G-J, making it less informative. Move it to the SI and show only few examples showing the trend. Otherwise, it is overshoot. Another option is to show in the figure only the calculated critical concentration and show the raw data at the SI.

2. It is highly recommended to have denaturation experiment using dynamic scanning calorimetry (DSC) with and without the inhibitors. This can show from thermodynamic point of view the thermal stability. Another option is to have temperature dependent secondary structure characterization (can be

performed with circular dichroism, SAXS or FTIR) showing how the structure changes with and without the inhibitors.

3. The aggregation experiments (Figure 3) were performed in relatively high pH. Why? Isn't it better to measure it in physiological conditions and more conventional buffer (or even simulated body fluids, SBF solution)? Moreover, the higher pH can affect the ThT fluorescence and aggregation in general (see "Revisiting thioflavin T (ThT) fluorescence as a marker of protein fibrillation – The prominent role of electrostatic interactions", JCIS 2020).

4. One control result that is quite important is the β -lactamase activity in vitro, before aggregation, after aggregation, with and without the inhibitors. I suggest to perform kinetic assay showing the decrease in the catalytic activity in β -lactam degradation. There are several β -lactam chromophores that are used as surrogates for these assays, for example nitrocefin degradation assay. Although I don't think this is necessary for all the inhibitors, showing how the catalytic efficiency of β lactamases is changing due to aggregation etc., might be important and significant.

Minor revisions:

1. At the abstract and the introduction, the full meaning of TEM/SHV should be added before using it, as it is not clear at the very beginning (for example for TM it appears only at line 69 but it is used earlier).

2. The abstract uses several terms/nomenclatures that are not so common for the audience. It is better to re-write the abstract to better clarify this and simplify it for readers that are more general.

3. A short description of the mode-of-action of β -lactamases should be added to the introduction (describing shortly the nucleophilic attack and the chemical circumstances leading to activity at the active site).

4. Either the last paragraph of the introduction or the first paragraph of the result section should emphasize the strategy of this experimental approach. The ending paragraph of the introduction should be widened and tell the reader what is going to appear in the next part [I think it'll help the reader to better understand the whole experimental process before getting into details].

5. A scheme describing the experimental approach, and specifically the steps in which the inhibitors were designed (and the sub groups in the dataset) can help the reader to track the whole process.

6. Figure 1 – the text in the figure (the axis titles) are not in the same font/size.

Reviewer #3 (Remarks to the Author):

Title: Exploiting the aggregation propensity of beta-lactamases to design inhibitors that induce enzyme misfolding

The review manuscript entitled "Exploiting the aggregation propensity of beta-lactamases to design inhibitors that induce enzyme misfolding" intends to develop synthetic peptides designed to exploit the structural defects of beta lactamases by inducing their misfolding into intracellular inclusion bodies. Manuscript needs improvement and some suggestions were here providing in order to improve the manuscript quality.

Suggestions

1. The methods of bioinformatic analyses are poorly described here. Authors are invited to better explain the logic behind peptide discovery. It is unclear how did authors reach to a positive peptide design. Otherwise, authors probably have some negative results in this specific field and could be important if authors could add a briefly discussion about negative peptides.
2. Authors must present the purity degree for peptides used. Please add mass spectrometry analyses of all of them at supplementary material.
3. Structural studies of peptides are not described in methods. Moreover, molecular dynamics studies could benefit a better understanding of mechanism of action as well complex stability. Please perform it.
4. At in vitro checkerboard analyses, authors must add the FICI values for all combinations tested. Please add it.
5. At in vivo analyses, authors must also present the mortality rate decrease in addition to CFU counts. Please add such information.

Point-by-point response

Reviewer #1:

The authors aim to show the efficacy of the TEM3.2 in the treatment of UTI, to eliminate bacterial growth from the urinary tract. As suggested, I have focused on the mouse model and the rest I have only glanced at. Therefore, misunderstandings might have happened. The mouse model is commonly used by us and others. One can of course argue that more sophisticated analysis could have been done, but I think the only thing that was relevant to them were the bacterial count. Still, I have some comments on the experimental and associated parts. Please see below.

Results:

Line 373: The authors comment different cell-lines used to show that the peptide is not cytotoxic. A wide range of cells have been tested. However, I have problems understanding why these specific cells were chosen and the immediate relevance for their study. The cells were derived from i.e. cervical cancer, distal lung epithelial cells and neuroblastoma. I lack bladder cells.

Modelling toxicity in cell lines is of course only a first step and, with these very early molecules, was not intended to be comprehensive in any way. We have now included cytotoxicity data on a bladder cell line (HT-1376) in the supplementary materials (Sup Fig 10). However, we left the other data since we do not want to focus solely on bladder cell toxicity because the peptide described are administered systemically, so toxicity could occur anywhere, and the peptides can potentially be used for infection of other tissues (our design is not bladder-specific).

Line 378: The authors mention that they treat cell lines, but it is not stated which cells and unfortunately, I am therefore not sure if it really is the HeLa cells as indicated in the Figure legends. I would suggest using a bladder cell line instead, since this section is about UTI.

The co-culture was indeed performed with HeLa cells we have clarified this in the text. As mentioned above, we have deliberately not focused on bladder cells. The paper is not exclusively about UTI, the model is in there to illustrate therapeutical potential, but we do not make major breakthroughs towards treating UTI. The co-culture studies were not intended to validate the use in the UTI model and were conducted using the Hela cell line, as was correctly indicated in the figure legend.

Line 386: The dose response curve mentioned in the text would fit well in the Supplement and would improve the understanding for the reader.

These data have now been included in supplementary table 4.

Line 396: Were mice inoculated with both E coli blaTEM-1 and at the same time either tazobactam or TEM3.2?

Tazobactam or peptides were administered one hour after infection. We included this in the text or in the Material and method section to make it easier to understand.

Line 396: At 60 and 120 min respectively mice were treated with ampicillin + FITC TEM32 (? Is it a typo and should instead be TEM3.2?) given either IV, IP or SC and as control tazobactam. To pretreat at the same time as start of infection is often done but is unfortunately far from real life. I would therefore recommend treating only after 60 and 120 min, especially as the authors claim it could be a therapeutic alternative (Line 409).

FITC-TEM32 has been corrected to FITC-TEM3.2

Line 397 already indicated that we treated the animal 60 and 120 minutes **after** inoculation, exactly as the reviewer suggests. To improve clarity, the term "post-inoculation" was changed to "post-infection" and an experimental scheme was added to the figure.

Figure 4: The labeling has drifted and should be adjusted. The figure legend is wrong and indicates that the bacterial growth is shown in figures C,D,E, but should be D,E,F.

We have corrected this.

Figure 4. Only 4 mice were included in each group. Although there are significances, I suggest adding a few more mice in each group to secure the results, since there is sometimes a big variation in results.

We have now increased the number of animals to 12, there is indeed spread, but the conclusions remain unchanged.

Figure 4. It might be obvious to everyone, except for me, but what is LOD as written in the figure 4 D,E,F

LOD stands for limit of detection. We have removed the lines to avoid confusion.

Reviewer #2:

Review:

The manuscript entitled "Exploiting the aggregation propensity of beta-lactamases to design

inhibitors that induce enzyme misfolding” (NCOMMS-23-00655) by Schymkowitz, Rousseau and others focuses on the aggregation of β -lactamases, a group of enzymes that degrade β -lactam antibiotics and considered the major contributors for antibiotic resistance. This work elucidates the effect of β -lactamases aggregation on their activity and resistance to antibiotics. The paper is interesting and important, both from basic perspective and its applications in fighting biofilm resistance to antibiotics. Yet, the manuscript should be edited to better clarify the take-home messages and to explain it to a wider audience. In addition, several more controls are needed to elaborate on the chemical mechanism behind the described phenomenon and inhibitors. Most importantly, few of the results appearing in the figures should be removed to the supplemental section, in order to keep the manuscript focused and clear. Specific comments:

Major revisions:

1. Figure 1 – the figure is too dense and contains a lot of data that is not crucial for understanding the main message and can be removed to the supplementary information section (for example, Figure 1D is nice and impressive but not necessary in the figure and should be only mentioned in the text). Also, too many concentrations are showed at Figure 1G-J, making it less informative. Move it to the SI and show only few examples showing the trend. Otherwise, it is overshoot. Another option is to show in the figure only the calculated critical concentration and show the raw data at the SI.

Figure 1 and supplementary Figure 1 have been reworked to reflect the reviewer’s suggestions, while still showing the full dataset.

2. It is highly recommended to have denaturation experiment using dynamic scanning calorimetry (DSC) with and without the inhibitors. This can show from thermodynamic point of view the thermal stability. Another option is to have temperature dependent secondary structure characterization (can be performed with circular dichroism, SAXS or FTIR) showing how the structure changes with and without the inhibitors.

To satisfy the reviewer’s comment, we have performed urea denaturation in the presence of a molar excess of tazobactam and compared it to the condition without. This revealed a two-state denaturation of the protein with a stable intermediate. The difference between the curves is subtle, but the fitting indicated that one of the first transition (from Native to Intermediate) may be affected by the presence of the inhibitor, but the effect is not very convincing on the basis of this experiment alone and would require more additional work, which lead us beyond the scope of the current study. Therefore, we provide the data here for reviewer’s sake and hope he will agree that our current manuscript is already very dense in data, with much of it in supplementary materials and that these insights do not alter the main conclusions of the current manuscript.

Figure 1: Chemical denaturation of TEM-1 protein, in presence (left) and absence (right) of a molar excess of tazobactam. The line shows the fit to a 3-state equilibrium denaturation model. The plots show a single repeat.

Table 1: Results from fitting 3 replicates

	$\Delta G1$ (kcal/mol)	$\Delta G2$ (kcal/mol)	Sum (kcal/mol)
no tazo	5.40 \pm 0.37	5.00 \pm 0.14	10.40
with tazo	6.25 \pm 0.66	4.03 \pm 0.24	10.28

In addition, we also tried the FTIR experiment, both at 30C and in a temperature ramp, but although we observe differences, we feel they do not provide a clear-cut view on the mechanism. We therefore provide the data here for the reviewer, but have not included them in the manuscript.

Figure 2: Fourier-Transform Infrared Spectroscopy (FTIR) of TEM-1 protein, in presence (orange) and absence (blue) of a molar excess of tazobactam. Left shows the amide I and II bands of the spectra at 30C, right shows the temperature dependence of the peak at 1626 cm⁻¹ (biggest difference on the plot on the left), normalized to 1650 cm⁻¹.

3. The aggregation experiments (Figure 3) were performed in relatively high pH. Why? Isn't it better to measure it in physiological conditions and more conventional buffer (or even simulated body fluids, SBF solution)? Moreover, the higher pH can affect the ThT

fluorescence and aggregation in general (see “Revisiting thioflavin T (ThT) fluorescence as a marker of protein fibrillation – The prominent role of electrostatic interactions”, JCIS 2020).

We are aware of the potential electrostatic effects, and in order to control for this we typically compare experiments using ThT to ones using the oppositely charged pFTAA. Since both experiments agree, we conclude that the result is not driven by the electrostatics. Moreover, as the poly-phosphate effect clearly shows, in the full matrix many components have an effect and they may combine in non-linear ways. Therefore, we interpret the *in vitro* experiments only to confirm the aggregation propensity of the peptides, and then we shifted indeed to in-cell experiments for the rest of the figure (similar to what the reviewer suggests), which is indeed the condition that matters most.

4. One control result that is quite important is the β -lactamase activity *in vitro*, before aggregation, after aggregation, with and without the inhibitors. I suggest to perform kinetic assay showing the decrease in the catalytic activity in β -lactam degradation. There are several β -lactam chromophores that are used as surrogates for these assays, for example nitrocefin degradation assay. Although I don't think this is necessary for all the inhibitors, showing how the catalytic efficiency of β -lactamases is changing due to aggregation etc., might be important and significant.

We agree with this suggestion and have now performed additional experiments to show inactivation of the enzyme upon co-incubation with the peptin, based on nitrocefin. It was added as supplementary Figure 8 and is mentioned in the text describing Figure 3.

Minor revisions:

1. At the abstract and the introduction, the full meaning of TEM/SHV should be added before using it, as it is not clear at the very beginning (for example for TM it appears only at line 69 but it is used earlier).

We have altered the flow of abstract and introduction slightly to resolve this issue. Thank you for pointing it out.

2. The abstract uses several terms/nomenclatures that are not so common for the audience. It is better to re-write the abstract to better clarify this and simplify it for readers that are more general.

We have reworked the abstract to improve readability, but the strict character limit does preclude elaborating on basic concepts.

3. A short description of the mode-of-action of β -lactamases should be added to the introduction (describing shortly the nucleophilic attack and the chemical circumstances leading to activity at the active site).

A brief statement has been added to the introduction.

4. Either the last paragraph of the introduction or the first paragraph of the result section should emphasize the strategy of this experimental approach. The ending paragraph of the introduction should be widened and tell the reader what is going to appear in the next part [I think it'll help the reader to better understand the whole experimental process before getting into details].

A paragraph in this vein was added at the end of the introduction.

5. A scheme describing the experimental approach, and specifically the steps in which the inhibitors were designed (and the sub groups in the dataset) can help the reader to track the whole process.

A schematic has been added to the supplementary materials.

6. Figure 1 – the text in the figure (the axis titles) are not in the same font/size.

This has been corrected as much as possible, given the diversity of plots.

Reviewer #3:

Title: Exploiting the aggregation propensity of beta-lactamases to design inhibitors that induce enzyme misfolding

The review manuscript entitled “Exploiting the aggregation propensity of beta-lactamases to design inhibitors that induce enzyme misfolding “ intends to develop synthetic peptides designed to exploit the structural defects of beta lactamases by inducing their misfolding into intracellular inclusion bodies. Manuscript needs improvement and some suggestions were here providing in order to improve the manuscript quality.

Suggestions

1. The methods of bioinformatic analyses are poorly described here. Authors are invited to better explain the logic behind peptide discovery. It is unclear how did authors reach to a positive peptide design. Otherwise, authors probably have some negative results in this specific field and could be important if authors could add a briefly discussion about negative peptides.

The details of the bioinformatics analyses were supplied in materials and methods, we have rearranged these to improve clarity.

In this work, we did not explore other charged amino acids, we just applied the previously developed design paradigm for antibacterial peptins, which we had good efficacy and uptake in previous studies with other antibacterials. To clarify this, we have now added the correct citations for this, which were missing.

“ Previous work has shown that good bacterial uptake can be achieved using one positively charged arginine residue on the N-terminal side of each APR in the tandem and two arginine residues on the C-terminal side^{1,2}. “

2. Authors must present the purity degree for peptides used. Please add mass spectrometry analyses of all of them at supplementary material.

Mass spectrometry and HPLC analysis have been added in supplement.

3. Structural studies of peptides are not described in methods. Moreover, molecular dynamics studies could benefit a better understanding of mechanism of action as well complex stability. Please perform it.

After consulting with several MD experts, we respectfully decline to engage in MD simulations of the peptide. The paper has a strong experimental focus and, based on our discussions, it is very questionable if the type of MD that could reasonably be done within the framework of this revision would have any impact at all on the conclusions of our study.

For the benefit of the reviewer, we are engaging in comprehensive studies in this direction, which we anticipate will take several years more work.

4. At in vitro checkerboard analyses, authors must add the FICI values for all combinations tested. Please add it.

A new table with FICI values has been added to the supplementary data (Supplementary Table 3).

5. At in vivo analyses, authors must also present the mortality rate decrease in addition to CFU counts. Please add such information.

For ethical reason, in the current animal model we used a bacterial load (10^8 bacteria) that produces a robust urinary tract infection, but at this early time point does not kill the animals, therefore we measure the CFU in Infected Tissue. The mortality rate is hence zero.

References

- 1 Bednarska, N. G. *et al.* Protein aggregation as an antibiotic design strategy. *Mol Microbiol* (2015). <https://doi.org:10.1111/mmi.13269>
- 2 Khodaparast, L. *et al.* Aggregating sequences that occur in many proteins constitute weak spots of bacterial proteostasis. *Nature communications* **9**, 866 (2018). <https://doi.org:10.1038/s41467-018-03131-0>

REVIEWERS' COMMENTS

Reviewer #2 (Remarks to the Author):

The authors addressed my comments

Reviewer #3 (Remarks to the Author):

The manuscript was revised and it is acceptable at present form

Point-by-point response

Reviewer #2:

The authors addressed my comments

Reviewer #3:

The manuscript was revised and it is acceptable at present form

We thank the reviewers for their work.